# A tunable transition metal dichalcogenide entangled photon-pair source

Maximilian A. Weissflog [1,2] ✉, Anna Fedotova [1,3], Yilin Tang[4], Elkin A. Santos[1], Benjamin Laudert[1], Saniya Shinde [1], Fatemeh Abtahi[1], Mina Afsharnia[1], Inmaculada Pérez Pérez [1], Sebastian Ritter [1,2,5], Hao Qin[4], Jiri Janousek[4,6], Sai Shradha[1,7], Isabelle Staude [1,3], Sina Saravi [1], Thomas Pertsch [1,5], Frank Setzpfandt [1,5], Yuerui Lu [4,6] ✉ & Falk Eilenberger [1,2,5] ✉

Entangled photon-pair sources are at the core of quantum applications like quantum key distribution, sensing, and imaging. Operation in space-limited and adverse environments such as in satellite-based and mobile communication requires robust entanglement sources with minimal size and weight requirements. Here, we meet this challenge by realizing a cubic micrometer scale entangled photon-pair source in a 3R-stacked transition metal dichalcogenide crystal. Its crystal symmetry enables the generation of polarization-entangled Bell states without additional components and provides tunability by simple control of the pump polarization. Remarkably, generation rate and state tuning are decoupled, leading to equal generation efficiency and no loss of entanglement. Combining transition metal dichalcogenides with monolithic cavities and integrated photonic circuitry or using quasi-phasematching opens the gate towards ultrasmall and scalable quantum devices.

Entangled photon pairs are the key-enabler for real-world implementations of quantum technologies like secure quantum key distribution[1,2], quantum sensing, and imaging[3], as well as distributed quantum computing schemes[4]. Consequently, a large variety of entangled photon-pair sources (EPS) has been developed, often relying on spontaneous parametric down-conversion (SPDC) in second-order nonlinear crystals[5]. Setting out from the first EPS implementations based on single bulk crystals[6], ever more complex source designs were developed to meet requirements for the degree of entanglement, quantum state fidelity, tunability, and brightness of the sources. Solutions to create entangled photon pairs are typically based on the interference of two distinct SPDC processes and range from using two crossed nonlinear crystals[7] via a combination of different down-conversion paths in Sagnac and linear interferometers[5] to integrated photonic systems[8]. Achieving a high degree of entanglement in such sources imposes very narrow tolerances on the properties of the different SPDC processes to allow the necessary coherent superposition. This technological challenge is further increased by the demand to operate these complex sources in adverse and space-limited environments, such as a satellite[9,10], or in customer-level applications like mobile-phone quantum key distribution, which also need a simple and scalable approach. Tunability between different entangled states is desirable for active quantum networks[11]. In light of these demands, a requirement list for an ideal EPS design would be the generation of high-fidelity, maximally entangled (Bell) states, switching between different entangled states, wide frequency coverage, and

[1]Institute of Applied Physics, Abbe Center of Photonics, Friedrich Schiller University Jena, Albert-Einstein-Straße 15, Jena 07745, Germany. [2]Max Planck School of Photonics, Hans-Knöll-Straße 1, Jena 07745, Germany. [3]Institute of Solid State Physics, Friedrich Schiller University Jena, Helmholtzweg 3, Jena 07743, Germany. [4]School of Engineering, College of Science and Computer Science, The Australian National University, Canberra, ACT, Australia. [5]Fraunhofer Institute for Applied Optics and Precision Engineering IOF, Albert-Einstein-Straße 7, Jena 07745, Germany. [6]Australian Research Council Centre of Excellence for Quantum Computation and Communication Technology, The Australian National University, Canberra, ACT, Australia. [7]Institute for Condensed Matter Physics, Technical University of Darmstadt, Hochschulstraße. 6-8, Darmstadt 64289, Germany. ✉e-mail: maximilian.weissflog@uni-jena.de; yuerui.lu@anu.edu.au; falk.eilenberger@uni-jena.de

high brightness, combined with a robust, scalable design and small footprint using as few optical components as possible. A very promising nonlinear material for such an EPS are 3R-phase transition metal dichalcogenide (3R-TMD) crystals, for instance, 3R-phase molybdenum disulfide (3R-MoS$_2$). Owing to its bulk-noncentrosymmetry, the signal yield of nonlinear conversion in multilayer 3R-MoS$_2$[12–15] is drastically increased compared to monolayer (ML) TMDs[16–20]. The high second-order nonlinearity of 3R-MoS$_2$ ($\chi^{(2)} \approx 100$ pm/V [21] in transparency region and peak values $\chi^{(2)} > 800$ pm/V[14,21,22] with excitonic enhancement in absorbing region) is at the same level or largely exceeds the nonlinearity of many established materials (beta barium borate (BBO) $\chi^{(2)} = 3.9$ pm/V[23], potassium titanyl phosphate (KTP) $\chi^{(2)} = 29.2$ pm/V, lithium niobate (LiNbO$_3$) $\chi^{(2)} = 49.8$ pm/V, gallium arsenide (GaAs) $\chi^{(2)} = 340$ pm/V (absorbing) and gallium phosphide (GaP) $\chi^{(2)} = 141$ pm/V all at fundamental wavelength 1064 nm ref. [24]). Leveraging this, it was demonstrated with second-harmonic generation (SHG) that 3R-MoS$_2$ requires two orders of magnitude shorter propagation length to reach the same second-order nonlinear conversion efficiency in the telecom range as LiNbO$_3$[13]. While in this case, crystals of only one coherence length were compared, the conversion efficiency of 3R-MoS$_2$-based nonlinear sources can be scaled to the required level, e.g. through quasi-phasematching. Similar to periodic poling in ferroelectric materials, the nonlinearity in stacks of several multilayer 3R-MoS$_2$ crystals can be periodically modulated by suitably rotating consecutive crystals[20], which was experimentally demonstrated very recently[21].

In this work, we show that 3R-TMDs can serve as the core component of a compact and highly tunable EPS by demonstrating the generation of maximally entangled photon pairs from submicron 3R-MoS$_2$ crystals. Our photon-pair source based on 3R-MoS$_2$ leverages the crystal symmetry of this van-der-Waals material to intrinsically create polarization entanglement. We demonstrate the broadband generation of maximally polarization entangled Bell states with a measured fidelity of up to 96%. The need for external optical elements to create entanglement is obliterated, allowing to keep the optical system as simple as possible. Remarkably, the output quantum state of the TMD crystal can be easily tuned to different Bell and other maximally entangled states, all with the same generation efficiency. This property fundamentally stems from the crystal symmetry and goes beyond other recent demonstrations of thin-film nonlinear sources[25–27]. The high photoluminescence present in monolayer (ML) TMDs[28] is suppressed in 3R-MoS$_2$, which is a decisive advantage compared to previous, inconclusive attempts to photon-pair generation in ML-TMDs[29,30]. The focus of this work is the demonstration of fundamental properties of generating entangled quantum states in 3R-MoS$_2$ and not yet a highly efficient and integrated device design. For specific technological applications requiring a high brightness of photon pairs in defined spectral bands[10], orders of magnitude enhancement of the pair rate in the desired range may be achieved by periodic poling or integrating the nonlinear TMD crystal into singly- or doubly resonant, monolithic cavities[31,32], an available technological process[33,34]. Based on our work, these readily developed technologies can, in the future, be combined to realize highly compact, flexible, and robust entangled photon-pair sources based on TMDs.

## Results

### Fundamentals of photon-pair generation and polarization entanglement in transition metal dichalcogenides

In the monolayer limit, TMDs with the structural form $MX_2$ ($M$ = Mo,W; $X$ = S,Se) are non-centrosymmetric and have a crystal lattice with three-fold rotational symmetry around the $z$-axis, corresponding to the point group $D_{3h}$. This leads to a $\hat{\chi}^{(2)}$ nonlinear tensor with non-vanishing elements $\chi^{(2)}_{\alpha\beta\gamma} = \chi^{(2)}_{yyy} = -\chi^{(2)}_{yxx} = -\chi^{(2)}_{xxy} = -\chi^{(2)}_{xyx}$ [35]. The $x$- and $y$-direction are defined based on the crystallographic zigzag (ZZ) and armchair (AC) directions, see Fig. 1b. This nonlinear tensor couples electric fields

with signal and idler frequencies $\omega_s$, $\omega_i$, and polarization indices $\alpha, \beta$ to a higher-frequency pump field with $\omega_p = \omega_s + \omega_i$ and polarization index $\gamma$. This enables classical three-wave mixing processes like second-harmonic generation (SHG) and sum-frequency generation (SFG) in TMDs, which were extensively studied[16–18]. The same nonlinearity also enables SPDC, where, due to vacuum fluctuations, pump photons with frequency $\omega_p$ spontaneously split into pairs of signal and idler photons with frequencies $\omega_s$ and $\omega_i$.

So far SPDC, which is the reverse process of SFG, could not be observed in TMDs[29,30]. Using TMDs for SPDC would be particularly interesting since their nonlinear tensor ensures that the generated signal and idler photons are intrinsically polarization-entangled. To demonstrate this, let us first consider a $y$-polarized pump photon. In this case, two pathways for down-conversion exist simultaneously, namely $|y\rangle_{\text{pump}} \xrightarrow{\chi^{(2)}_{yyy}} |yy\rangle$ and $|y\rangle_{\text{pump}} \xrightarrow{\chi^{(2)}_{xxy}} |xx\rangle$. Since both processes are coherently driven by the same pump photon, the ensuing quantum state is a coherent superposition of the two conversion possibilities with equal magnitudes as $\chi^{(2)}_{yyy} = -\chi^{(2)}_{xxy}$. The resulting polarization quantum state is $|\Phi^-\rangle = 1/\sqrt{2}(|xx\rangle - |yy\rangle)$. This is one of the Bell states, a maximally entangled quantum state with high importance in quantum information processing. Equivalently, an $x$-polarized excitation results in a coherent superposition of the two down-conversion paths $|x\rangle_{\text{pump}} \xrightarrow{\chi^{(2)}_{yxx}} |xy\rangle$ and $|x\rangle_{\text{pump}} \xrightarrow{\chi^{(2)}_{yxx}} |yx\rangle$. This generates the Bell state $|\Psi^+\rangle = 1/\sqrt{2}(|xy\rangle + |yx\rangle)$, again maximally entangled. For a pump polarization rotated by the angle $\varphi_p$ with respect to the $x$-axis, the generated state is a superposition of these two Bell states in the form

$$|\psi\rangle = \frac{\sin(\varphi_p)}{\sqrt{2}}(|HH\rangle - |VV\rangle) + \frac{\cos(\varphi_p)}{\sqrt{2}}(|HV\rangle + |VH\rangle), \qquad (1)$$

where we have used now the horizontal $|H\rangle$ and vertical $|V\rangle$ basis states in the far-field for the notation. These are co-aligned with the crystallographic $x$-axis (ZZ) and the $y$-axis (AC), respectively (see Fig. 1b). Based on this general form of the quantum state, it is straightforward to characterize the entanglement of states that lie in between the $\Psi^+$-state for $x$-polarized excitation ($\varphi_p = 0°$, horizontal) and the $\Phi^-$-state for $y$-polarized excitation ($\varphi_p = 90°$, vertical). As an entanglement measure, we employ the concurrence $C$, a quantity ranging between $C = 0$ for separable and $C = 1$ for fully entangled states[36]. In Fig. 1c, we plot the fidelity of the general state Eq. (1) with the Bell states $\Psi^+$- and $\Phi^-$ as well as the concurrence $C$ for a full rotation of the pump polarization angle $\varphi_p$. For a full derivation and the used definitions of concurrence and fidelity, refer to Supplementary Note 1. While the state fidelities for the two Bell states peak at $\varphi_p = 0°$ and 90°, the concurrence is $C = 1$ for all $\varphi_p$. In fact, the output polarization state from the TMD for any pump angle is always maximally entangled. Furthermore, analogous to the case of classical frequency up-conversion[17], due to their crystal symmetry, the spontaneous down-conversion rate in TMDs is independent of the pump polarization. Therefore, TMDs generate fully entangled polarization states that are tunable with constant efficiency by means of pump polarization change.

A drawback of ML-TMDs is the low absolute signal yield in nonlinear conversion due to the very small interaction length with the medium[13]. More promising for the practical implementation of nonlinear devices based on TMDs is the use of moderately thicker crystals, with a stacking scheme that still preserves non-centrosymmetry. One such material is the 3R-polytype of TMDs like MoS$_2$[12–15], where the layer-stacks are arranged in an ABC-ABC scheme that has no inversion center (one stacking period consists of three layers, compare inset of Fig. 1a)[12].

Since 3R-MoS$_2$ maintains the 3-fold rotational crystal symmetry and the related in-plane nonlinear tensor elements, it belongs to the $C_{3v}$ point group, also the thicker 3R-crystal stacks are suited to

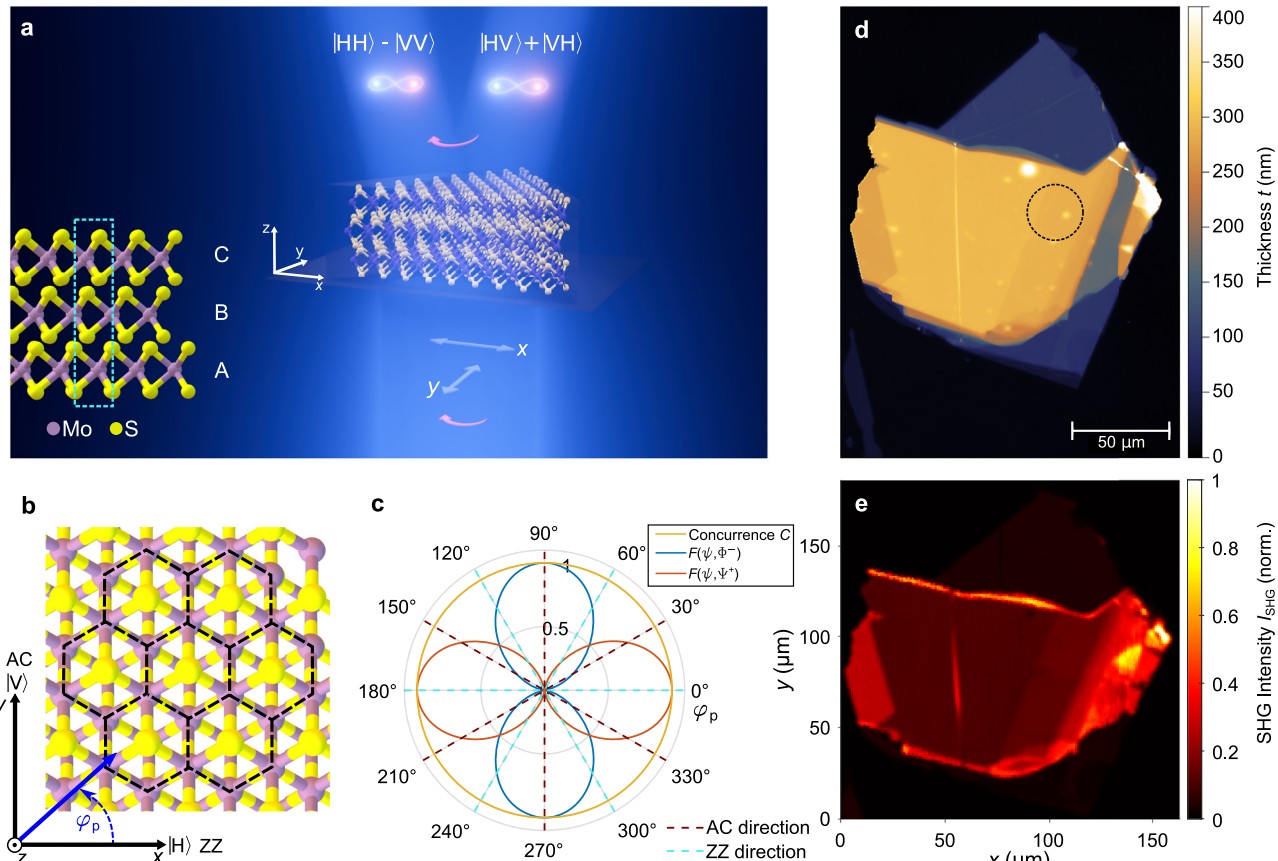

**Fig. 1 | Principle of entangled photon-pair generation in 3R-MoS₂.** **a** A multilayer 3R-stack of MoS₂ generates pairs of polarization-entangled signal and idler photons via spontaneous parametric down-conversion (SPDC). Depending on the orientation of the pump polarization, different maximally entangled Bell states are generated. Lower inset: a sketch of the ABC-stacking scheme in 3R-MoS₂. **b** Top view of the crystalline structure of 3R-MoS₂ stack shown in (**a**). The unit cells of the crystal are highlighted with black, dashed lines. The $x$- and $y$-coordinates, as well as the $|H\rangle$- and $|V\rangle$-polarization directions, are aligned with the zigzag (ZZ)- and armchair (AC)-directions of the crystal, respectively. The definition of the pump polarization angle $\varphi_p$ is marked in blue. **c** Theoretical evolution of the fidelities $F(\psi, \Phi^-)$ and $F(\psi, \Psi^+)$ of the polarization state $|\psi\rangle$ with the Bell states $|\Phi^-\rangle = 1/\sqrt{2}(|HH\rangle - |VV\rangle)$ and $|\Psi^+\rangle = 1/\sqrt{2}(|HV\rangle + |VH\rangle)$, respectively, as well as the concurrence $C$ for a full rotation of the pump angle $\varphi_p$. **d** Height-map of the investigated 3R-MoS₂ crystal obtained with a vertical scanning interferometer. The black circle marks the measurement region. **e** Map of second-harmonic generation (SHG) intensity for the 3R-MoS₂ crystal shown in (**d**), obtained for excitation at central wavelength $2 \times 788\,\text{nm} = 1576\,\text{nm}$.

generate polarization-entangled quantum states. The signal yield, however, is much higher than for a monolayer. The out-of-plane, $z$-polarized nonlinear tensor components of 3R-MoS₂ practically only contribute to the generated quantum state for very large collection numerical apertures, refer to Supplementary Note 2 for a detailed discussion.

**Experimental photon-pair generation and polarization analysis**
Experimentally, we aim for photon-pair generation in the technically relevant telecom band in the near infrared around $\lambda_{s,i} \approx 1550\,\text{nm}$. Using mechanical exfoliation, we fabricate a 3R-MoS₂ crystal with sub-wavelength thickness, see "Methods" section. In Fig. 1d, we show a height map of the crystal used as a photon-pair source in this work. For the SPDC measurement, we choose an area far away from the crystal edges and all cracks, which is important to minimize distortions of the nonlinear tensor induced by imperfections or strain[12,37]. To further define the measurement area for the SPDC experiments, we first spatially map the SHG emitted by the crystal, as shown in Fig. 1e. We choose the large area of 285 nm thickness, see the marking in Fig. 1d, which shows a strong SHG signal in the center of the crystal. The signal yield from this crystal exceeds an ML-MoS₂ by more than three orders of magnitude. We, however, note that this crystal thickness does not correspond to the global maximum conversion efficiency. This would be reached for a thickness of $t \approx 800\,\text{nm}$, close to the coherence length of

$L_c \approx 840\,\text{nm}$ for the SHG conversion process excited at 1576 nm. At this thickness, the SHG and, by correspondence, also SPDC efficiency would be further increased by a factor of $\approx 33\times$ (see Supplementary Fig. 5 and Supplementary Note 3). While we did not reach this optimum thickness with the limited control offered by mechanical exfoliation, nanofabrication techniques that allow precise thickness control and nanopatterning of (3R-)MoS₂ have already been demonstrated[22,38] and can be used for future sample fabrication. For the photon-pair measurements, we use an experimental setup with two fiber-coupled, time-correlated single-photon detectors, as shown in Fig. 2a. A pump beam with wavelength $\lambda_p = 788\,\text{nm}$ is focused onto the air-exposed side of the 3R-MoS₂ sample, and photon pairs are collected through the quartz substrate.

In our correlation experiment, any other emission from the sample in the same wavelength region would potentially mask the entangled photon signal. In particular, strong photoluminescence, as observed from direct bandgap transitions in ML-TMDs[28], could complicate the observation of photon pairs[30]. We measure photoluminescence from our sample under excitation at $\lambda_p = 788\,\text{nm}$ from the same pump laser as in the SPDC experiments. We observe no photoluminescence signal distinguishable from the detector dark-counts beyond $\lambda = 1300\,\text{nm}$ (see green shaded area Fig. 2b, pump intensity $\approx 1.67\,\text{MW cm}^{-2}$). This demonstrates the 3R-TMD's potential for low background photon-pair generation in the telecom wavelength band.

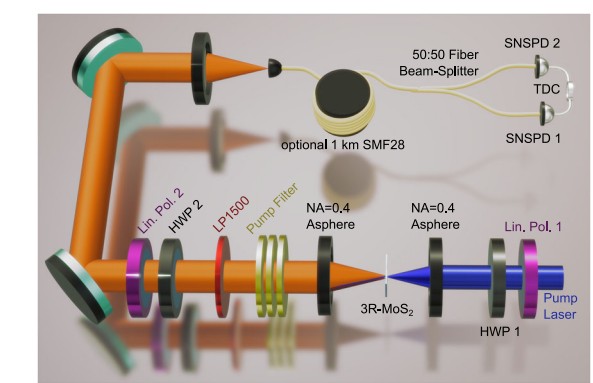

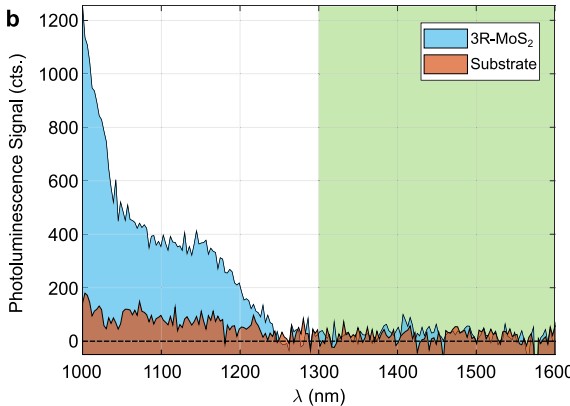

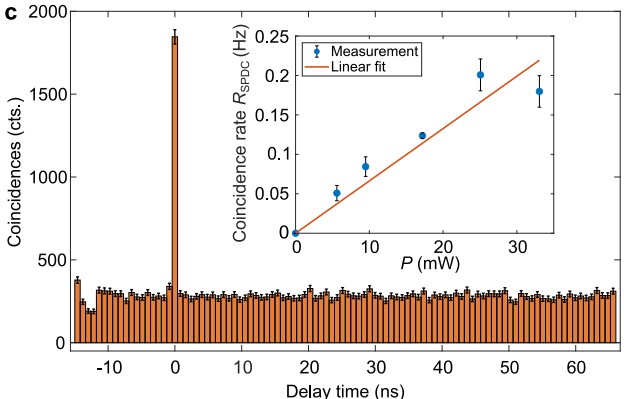

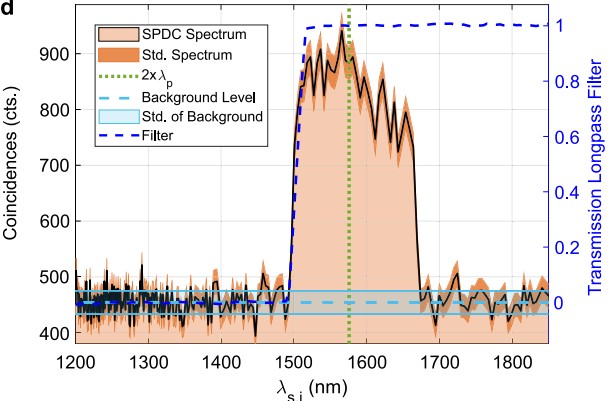

**Fig. 2 | Experimental observation of photon pairs from 3R-MoS₂. a** Hanbury Brown-Twiss interferometer for photon-pair correlation measurements: continuous wave diode laser at 788 nm, linear polarizer 1 (lin. pol.) and half-wave plate 1 (HWP) for setting pump polarization angle $\varphi_p$, aspheric lenses for focusing onto and collecting from the sample, long-pass interference filters (cut-on wavelength $\lambda = 1100$ nm) for pump suppression, optional HWP 2 and lin. pol. 2 as polarization analyzer with angle $\varphi_{pol}$, single mode fiber (SMF) and 50:50 fiber beamsplitter, superconducting nanowire single-photon detector (SNSPD), and time-to-digital converter (TDC). **b** Photoluminescence (PL) spectrum from 3R-MoS₂ crystal and SiO₂ substrate excited at $\lambda_p = 788$ nm. The green shaded area marks the region where no photoluminescence signal distinguishable from the detector darkcounts can be detected. **c** Coincidence histogram obtained from 3R-MoS₂ for excitation

with average power $P = 17.2$ mW and integration time 3.5 h, measured with a long-pass filter with cut-on wavelength 1500 nm. The coincidence-to-accidental ratio (CAR) in this measurement is CAR = 5.5. Error bars mark the standard deviation based on the Poissonian statistics of SPDC coincidence detection. Inset: Measured SPDC coincidence rate (blue dots) for different pump powers and its linear fit (orange line). **d** Spectrum of SPDC photons measured using fiber spectroscopy. The dashed green line marks the degenerate SPDC wavelength $\lambda_{deg} = 2 \times \lambda_p = 1576$ nm. The dark orange shaded areas mark the standard deviation based on the statistics of SPDC coincidence detection. The light blue lines show the level and standard deviation (std.) of the noise floor. The dashed, dark blue line is the transmission curve of the used long-pass filter that limits the SPDC spectrum.

Consequently, we perform pair-correlation measurements and observe a pronounced coincidence peak, compare Fig. 2c. After background subtraction, we measure $1563 \pm 43$ coincidence counts with a coincidence-to-accidental ratio (CAR) of CAR = $5.5 \pm 0.4$ for an integration time of 3.5 h and a pump power incident on the sample of 17.2 mW (pump intensity $\approx 648$ kW cm$^{-2}$). A normalized second-order correlation function $g^{(2)}(0) > 2$, related to CAR via CAR = $g^{(2)}(0)-1$[25], together with the linear scaling of the coincidence rate with pump power shown in Fig. 2c is clear evidence for the SPDC origin of the coincidence peak (see Supplementary Note 5A and raw data in Supplementary Fig. 8). The maximum CAR we observe is CAR = $8.9 \pm 5.5$ for a pump power of 5.6 mW (see Supplementary Fig. 9). Furthermore, we measure the SPDC spectrum using fiber spectroscopy[39], where the photon pairs are first sent through a long, dispersive medium before coincidence detection. In our experiment we insert 1 km of SMF28 single-mode fiber before the beamsplitter for the spectral measurement. After propagation through this medium with known dispersion, the arrival time difference between both photons can be mapped to their wavelength difference (see "Methods" for more details). As expected for a non-phase matched, thin crystal, the SPDC spectrum is very broad[25,40], compare Fig. 2d. In the experiment, the spectrum is limited only by the long-pass filter with cut-on 1500 nm, which is used

to limit the SPDC spectrum to the operating bandwidth of the fiber beamsplitter and for suppression of residual photoluminescence (filter curve shown as dashed, dark blue line in Fig. 2d).

The specific form of the nonlinear tensor of 3R-MoS₂ leads to a characteristic dependence of the generated photon's polarization on the pump polarization, which we characterize next. Here we first consider the two cases of either an unpolarized detection or a single polarizer for both SPDC photons. While this does not yet allow to measure the degree of polarization entanglement which we cover in the next section, the measurements with a single analyzer are particularly instructive to relate to the polarization dependence of SHG measurements in similar configurations[17]. As reference, we show in Fig. 3a a classical polarization-resolved second-harmonic measurement from 3R-MoS₂ observed through an analyzer, that is rotated in parallel to the pump polarization (see the "Methods" section). The characteristic six-fold symmetric pattern is oriented along the AC crystal direction (dashed brown line in Fig. 3a)[17]. For SPDC detection without a polarizer, we observe the expected constant coincidence rate, independent of the pump polarization (Fig. 3b)[19]. We assign the small fluctuations in the measured rate mainly to the polarization sensitivity of our SNSPD detectors, which in the telecom range is significant[41]. For measurements through a common analyzer for both

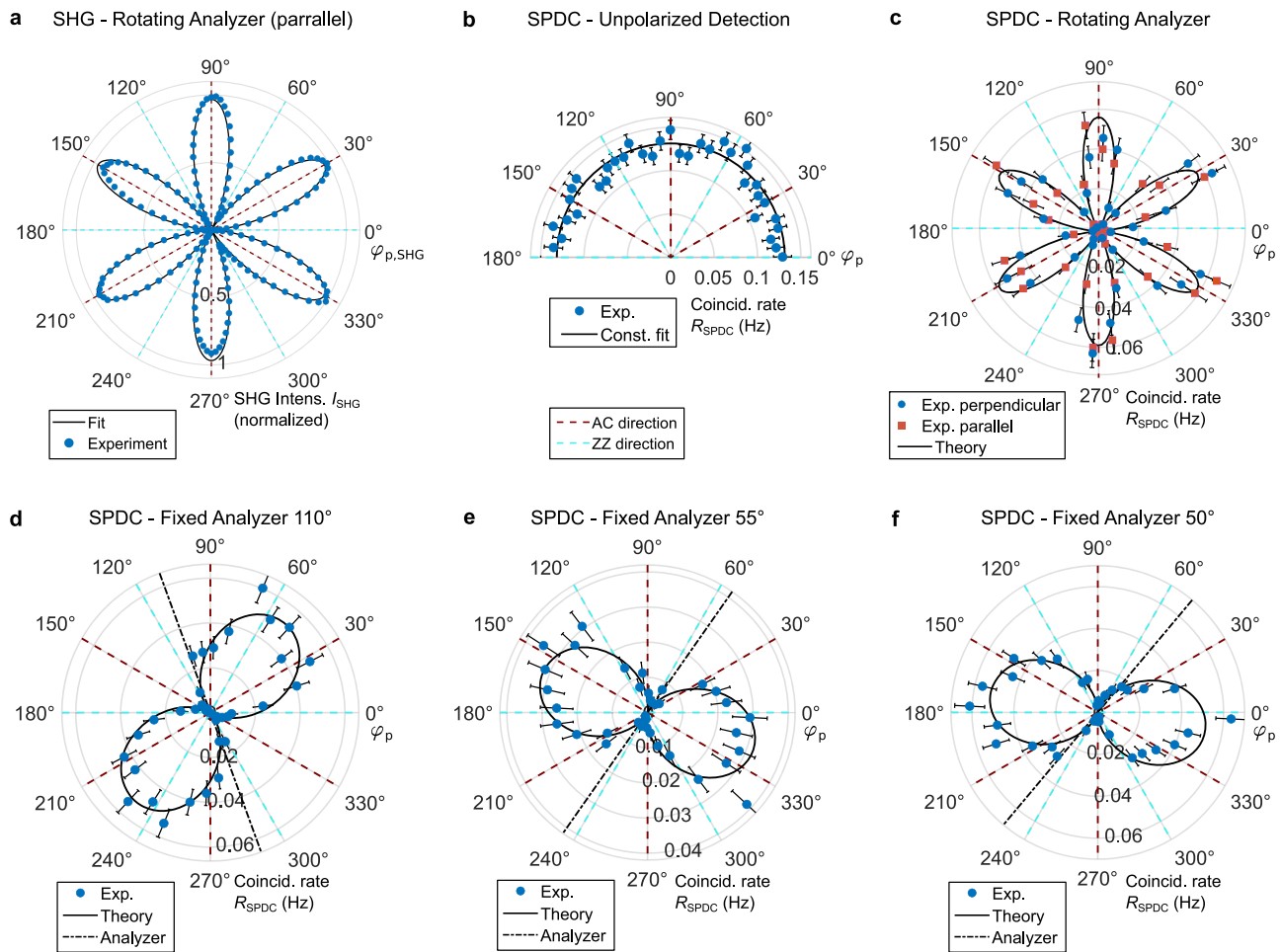

**Fig. 3 | Polarization analysis of photon pairs. a** Polarization-resolved second-harmonic measurement (blue circles) for the rotation of the pump polarization angle $\varphi_{p,SHG}$, and parallelly polarized, co-rotating analyzer. The black curve is a fit with the theoretically expected dependence $I_{SHG} \propto \sin^2(3\varphi_{p,SHG})$. The armchair (AC), and zigzag (ZZ) directions are marked with dashed brown and cyan lines, respectively. **b** Measured photon-pair rate for rotation of the SPDC pump polarization angle $\varphi_p$ for unpolarized detection. The black curve marks a fit with the theoretically expected constant function. **c** Measured photon-pair rate for rotating pump polarization $\varphi_p$ and detection through a co-rotating, perpendicularly oriented (blue circles), and parallelly oriented (purple squares) analyzer. In both cases, the expected $R_{SPDC} \propto \sin^2(3\varphi_p)$ dependence (black curve) is overlayed. **d–f** Measured photon-pair rates for the rotating pump polarization $\varphi_p$ and an analyzer fixed at **d** $\varphi_{pol} = 110°$ (20° offset from AC-axis), **e** $\varphi_{pol} = 55°$ (25° offset from AC-axis), and **f** $\varphi_{pol} = 50°$ (20° offset from AC-axis). The dash-dotted black line marks the direction of the analyzer transmission axis, the solid dashed line is the theoretically expected dependence $R_{SPDC} \propto \sin^2(2\varphi_{pol} + \varphi_p)$. Error bars mark the standard deviation based on the Poissonian statistics of SPDC coincidence detection.

signal and idler, the symmetry of the nonlinear tensor leads to a dependence of the SPDC rate $R_{SPDC}$ on pump polarization angle $\varphi_p$ and analyzer angle $\varphi_{pol}$ as $R_{SPDC} \propto \sin^2(2\varphi_{pol} + \varphi_p)$ (see Supplementary Note 1 for the derivation). To experimentally verify this, we insert an analyzer in front of the fiber and simultaneously rotate the pump and analyzer either in a parallel configuration $\varphi_{pol} = \varphi_p$ (orange squares in Fig. 3c) or perpendicular configuration $\varphi_{pol} = \varphi_p + \pi/2$ (blue dots in Fig. 3c). Both yield a characteristic six-fold, co-aligned pattern that matches the theoretically expected dependence of the form $R_{SPDC} \propto \sin^2(3\varphi_p)$ when analyzed in terms of the SPDC pump angle $\varphi_p$. Note that this is fully consistent with frequently reported SHG measurements that show a 30° shift between measurements with parallel or perpendicular polarizer angle[17], see Supplementary Note 1 for a more detailed discussion. Furthermore, by varying the pump polarization for several constant analyzer positions, we obtain a two-lobed pattern (see Fig. 3d–f), confirming the theoretically derived polarization dependence. In Supplementary Note 1 we also discuss the polarization dependence when keeping the pump polarization angle constant and only rotating the analyzer. The raw coincidence histograms for all results in Fig. 3 are found in Supplementary Figs. 10 and 11.

## Quantum-state tomography and Bell-state generation

To completely characterize the generated polarization quantum state and to prove entanglement between signal and idler photons, we perform a tomographic measurement in two mutually unbiased polarization bases[42]. To deterministically separate signal and idler photons, we insert a short-pass dichroic mirror with a cut-on wavelength of 1600 nm into the SPDC collection path, as depicted in Fig. 4a. The previously used long-pass with a cut-on wavelength of 1500 nm is removed now since the operating bandwidth of the dichroic mirror is larger than for the fiber beamsplitter. The broadband signal and idler spectra and the beamsplitter reflection spectrum are shown in Fig. 4b. Here, the width of the observed SPDC spectrum is mostly limited by the detection range of our experimental setup where the efficiency of the used SNSPDs slowly drops towards wavelengths much longer than their optimized operation wavelength at 1550 nm. Due to the correlation-based spectral measurement method, this simultaneously limits the short-wavelength side of the spectrum. Due to the collection in single-mode fibers and the use of an approximately non-polarizing dichroic mirror, the photon pairs remain indistinguishable in all degrees of freedom but their frequency. Using a combination of

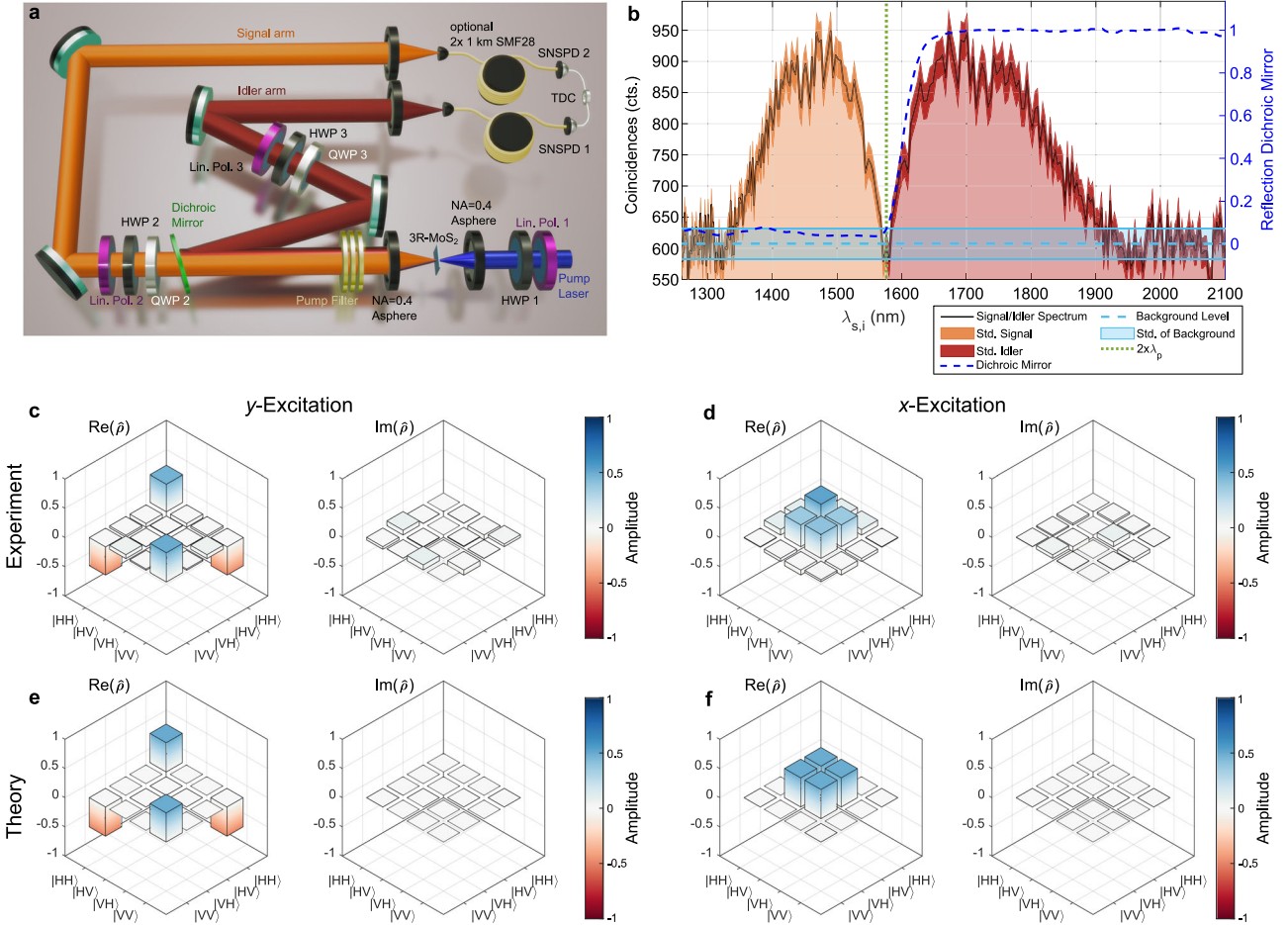

**Fig. 4 | Quantum-state tomography. a** Experimental setup for quantum polarization-state tomography. A short-pass dichroic mirror (DM) with cut-on wavelength at $\lambda = 1600$ nm splits the signal (orange, upper arm) and idler (red, lower arm) frequency modes. Quantum-state tomography is performed with a set of quarter-wave plate (QWP), half-wave plate (HWP) and linear polarizer in each arm, then temporal correlations are measured as with the experiment described in Fig. 2a. **b** Measured spectra of the signal (orange) and idler (red) frequency modes. The dip around the degenerate wavelength (dashed green line) is caused by the slight detuning of the cut-on wavelength of the dichroic mirror from the degenerate SPDC wavelength. The dark-shaded areas mark the statistical uncertainty. The light blue lines show the level and standard deviation of the noise floor. The dashed, dark blue line is the reflection curve of the dichroic mirror. **c** and **d** Experimentally measured polarization density matrices $\hat{\rho}$ for (**c**) for $y$-, and (**d**) $x$-polarized pump. For both cases, the real and imaginary parts Re($\hat{\rho}$) and Im($\hat{\rho}$), respectively, are shown. **e** and **f** Theoretically expected polarization density matrices obtained from fully vectorial Green's function calculations for $y$- and $x$-polarized excitations of 3R-MoS$_2$, respectively.

waveplates and a linear polarizer in both paths allows to set two arbitrary, independent polarization bases. By performing projections into 16 different basis states, the density matrix $\hat{\rho}$ of the polarization quantum state is fully determined[42] (see the "Methods" section). We use an established maximum-likelihood estimation method to determine a physically correct density matrix from measurements that are subject to noise and experimental uncertainties[42,43].

In Fig. 4c, d, we show the real and imaginary parts of the experimentally obtained density matrices $\hat{\rho}$ for a $y$- (c) and $x$-polarized (d) pump (raw data in Supplementary Note 5C and Supplementary Figs. 12–14). Additionally, we compute the theoretically expected state emitted from our 3R-MoS$_2$ with thickness $t = 285$ nm based on fully vectorial Green's function calculations, taking into account the realistic conditions in our experiment for pump focusing, collection NA, collected SPDC bandwidth, etc. (see the "Methods" section)[44,45]. Taking the ratio of in-plane and out-of-plane nonlinear tensor components measured in ref. 14 as a reference, these calculations predict the generation of ideal Bell states and compare the theoretical density matrices in Fig. 4e, f. This is closely matched by the experiment. For the $y$-polarized excitation, the measured density matrix has a fidelity of $F = 0.96$ with a $|\Phi^-\rangle = 1/\sqrt{2}(|HH\rangle - |VV\rangle)$ state and a concurrence of

$C = 0.973 \pm 0.002$, while for the $x$-polarized excitation, the fidelity with a $|\Psi^+\rangle = 1/\sqrt{2}(|HV\rangle + |VH\rangle)$ state is $F = 0.84$ and the concurrence $C = 0.82 \pm 0.02$.

## Discussion

In this work, we observe photon-pair generation via SPDC in a transition metal dichalcogenide. We chose 3R-MoS$_2$ for our demonstration because its strong nonlinearity is preserved in multi-layer stacks. Simultaneously, it is much less affected by photoluminescence than monolayer TMDs, which had prevented the observation of SPDC in prior experiments. We demonstrate that TMDs intrinsically generate maximally entangled polarization Bell states. Experimentally we show this for two different pump polarizations and then further theoretically derive that, in fact, for any linear pump polarization, a different maximally entangled state is generated while the generation efficiency is independent of the pump polarization. This decoupling of entangled state tuning from the generation efficiency results in a highly flexible and easy-to-operate, tunable entangled photon-pair source. Since all these properties are directly derived from the crystal symmetry, no external optical components like interferometers etc. are needed for generating entanglement. This is the simplest conceivable, tunable

entangled photon-pair source, a prerequisite for active quantum networks, which enable, for instance, multi-user quantum secret sharing[11]. Furthermore, this direct link to the crystal symmetry allows us to generalize our results for tuneable entanglement generation to other nonlinear materials of the same or similar symmetry group, for instance, lithium niobate or beta barium borate (BBO).

While we demonstrate here a prototype based on a single, thin 3R-MoS$_2$ crystal, the generation rate can be scaled to the required level, e.g. through quasi-phasematching. Similar to the periodic poling of ferroelectric nonlinear materials[5], the nonlinearity of 3R-MoS$_2$ can be periodically poled by stacking several multilayer crystals with appropriate rotation angles between consecutive crystals[20,21]. With this, quasi-phasematching between pump, signal, and idler waves is possible and the length of the 3R-TMD stack can be increased beyond one coherence length to match the photon-pair rate required in specific applications. Another way of scaling up the source brightness is cavity integration. A cavity resonance at the pump wavelength effectively extends the interaction length with the nonlinear crystal, drastically enhancing the total pair-generation rate while resonances at the signal and idler wavelength strongly increase the spectral brightness in the desired frequency bands[31,32]. The integration of TMDs into high-$Q$, monolithic cavities is a readily developed technology[33] with doubly-resonant cavities in reach[34]. Also, excitonic enhancement of the second-order nonlinear susceptibility is a promising avenue to further increase the source brightness[46]. Schemes like quasi-phasematching and cavity integration, which result in narrower spectral and spatial emission, would not only enhance the generation rate but also improve the photon-pair collection efficiency. Additionally, an enhanced SPDC efficiency, for instance, through longer interaction lengths, proportionally reduces the effect of photoluminescence[47]. This, together with the enhanced generation and collection efficiency, will improve the coincidence-to-accidental ratio in our source.

The demonstrated continuous tuning of the output state while maintaining maximal entanglement and a constant generation efficiency goes beyond what was shown with previously developed thin-film sources[26,27,40]. Combined with the avenues for scaling the generation rate, coincidence-to-accidental ratio, and pair collection efficiency, this gives TMDs a clear advantage as a nonlinear material platform for entangled photon-pair sources. Furthermore, the high refractive index of 3R-MoS$_2$ is well suited for strong field confinement when being nanostructured[22,38,48], making it a perfect platform for hyper-entangled photon-pair generation in resonant nanostructures[45,49,50]. Given that TMDs also withstand harsh conditions like those found in space[51] and can be easily integrated on top[52] or end-facet[53] of optical fibers[52], waveguides[54], and also metasurfaces[55], we expect to see their immediate use in microscale or integrated photonic circuits and entangled photon-pair sources. Combined with their extremely low requirements for size and weight with highly scalable fabrication routes, they will enable quantum communication and quantum sensing for medical applications, life sciences, the semiconductor industry, and consumer applications alike.

## Methods
### Sample fabrication
Bulk 3R-MoS$_2$ crystals were grown using the chemical-vapor transport technique[15]. Subsequently, 3R-MoS$_2$ flakes were prepared on poly-dimethylsiloxane (PDMS), which begins with mechanical exfoliation of the crystals. Afterward, the substrates were pre-treated by oxygen plasma in order to eliminate potential contamination and improve the adhesion, followed by a dry transfer method to transfer the 3R-MoS$_2$ flakes onto quartz substrates.

### Thickness characterization
Sample thicknesses were characterized by a surface profiler and vertical scanning interferometry (VSI, Bruker Contour GT-K). The surface

profiler and VSI are utilized to access the average thickness, surface roughness, and uniformity of the 3R-MoS$_2$ sample.

### Polarization-resolved SHG measurements and SHG mapping
Polarization-resolved SHG measurements were carried out with the same setup as used for quantum measurements but working in reverse: the fundamental beam was incident from one of the collecting fibers and focused/collected with the same optics (see Fig. 2a). As a laser source, a tunable femtosecond laser (Coherent Chameleon with optical parametric oscillator Angewandte Physik und Elektronik GmbH APE OPO-X) with pulse width 100 fs, repetition rate 80 MHz, at a central wavelength 1576 nm and with FWHM 10 nm was used. Note that the pulses were not sent through the normal detector fiber but through a shorter single-mode fiber (Thorlabs SMF-28-J9-CUSTOM) with a length of 0.5 m to avoid distortion of the pulses. The pump polarization was controlled with a half-wave plate (Thorlabs AHWP05M-1600), which rotated together with an analyzer placed in the collection path (Thorlabs WP25M-UB). Two short-pass filters (Thorlabs FELH850) installed in the collection path filtered out the fundamental beam, and SHG was detected with sCMOS camera (Excelitas pco.edge 4.2 bi), all not shown in Fig. 2a. For a detailed schematic of the experimental setup including the imaging arm, please refer to Supplementary Fig. 6. The detected polarization of the second-harmonic wave was kept parallel to the pump polarization creating a characteristic six-fold pattern. This measurement was used as a reference to identify the orientation of the AC and ZZ crystal directions in the 3R-MoS$_2$ sample.

SHG mapping was performed using a custom-built nonlinear microscopy setup. A fundamental beam from a tunable femtosecond laser (Spectra-Physics Mai Tai and optical parametric oscillator Inspire HF 100) with a pulse width of 100 fs, repetition rate 80 MHz, at a central wavelength of 1576 nm, and with FWHM 10 nm was focused onto the sample via a 20x NA = 0.4 objective (Mitutoyo). The polarization of the fundamental beam was fixed to be parallel to the AC-axis of 3R-MoS$_2$. The beam diameter reached <6 µm FWHM. The SHG signal was collected via a 100x NA = 0.85 objective (Zeiss) and passed through two short-pass filters to remove the fundamental beam. The sample was then scanned with 1 µm step-width on a motorized *XYZ*-stage (Newport M-VP-25XL-XYZR), while the second-harmonic signal was detected using an EMCCD camera (Andor, iXon3).

In both experiments, the excitation wavelength was chosen to correspond to the degenerate wavelength of SPDC pumped at $\lambda_p$ = 788 nm.

### Photon-pair correlation measurements
Photon-pair correlation measurements shown in Figs. 2 and 3 were performed using the home-built Hanbury Brown-Twiss interferometer outlined in Fig. 2a. A more detailed schematic of the experimental setup is provided in Supplementary Fig. 6. Excitation photons from a continuous-wave laser at $\lambda_p$ = 788 nm (diode laser, Thorlabs FPL785P) were sent through a linear polarizer and a half-wave plate for pump polarization control and focused onto the sample by an aspheric lens with numerical aperture NA = 0.4 (Thorlabs C110TMD-B), leading to a diffraction-limited $1/e^2$ pump beam radius of ≈1.3 µm. The measurement position is imaged in the experimental setup in a separate imaging arm via the same camera (Excelitas pco.edge 4.2 bi) as also used for SHG measurements. Compare Supplementary Fig. 6 for details. Subsequently, photon pairs were collected in transmission geometry using a similar lens with anti-reflection coating for the C-band (Thorlabs C110TMD-C). Pump photons were removed using three interference long-pass filters with cut-on wavelength 1100 nm (Thorlabs FELH1100). For measurements shown in Figs. 2 and 3, we also used a long-pass filter with a cut-on wavelength of 1500 nm (Thorlabs FELH1500) to suppress any residual photoluminescence and to limit the photon-pair bandwidth to the operation range of the fiber beamsplitter. The photon pairs were then coupled to single-mode

fibers (Corning SMF28), separated using a broadband fiber beamsplitter with central wavelength 1550 nm (Thorlabs TW1550R5F1), and directed to two superconducting single-photon detectors (SNSPD, Single Quantum Eos). Coincident detection events are registered with a time-correlator (qutools quTAG or ID Quantique ID800). For the polarization measurements with a common analyzer for both photons in Fig. 3, we implement a rotating analyzer using an achromatic half-wave plate (Thorlabs AHWP05M-1600) followed by a fixed linear polarizer (Thorlabs WP25M-UB), such that the polarization state in the detector fiber is always the same. This rules out the polarization dependence of the detectors. The total photon-pair detection efficiency of the setup $\eta_{\text{tot}}$ follows from $\eta_{\text{tot}} = T_{\text{opt}}^2 \times T_{\text{coupl}}^2 \times \eta_{\text{BS}} \times \eta_{\text{detec}}^2 \times \eta_{\text{LP}}^2 \approx 0.6\,\%$. For our setup, we estimate the following values: single photon optical transmission, including lenses, filters, mirrors, etc. $T_{\text{opt}} \approx 0.78$; single-mode fiber coupling efficiency $\eta_{\text{coupl}} \approx 0.35$; fiber beamsplitter non-uniformity and probabilistic splitting $\eta_{\text{BS}} \approx 0.95^2 \times 0.5 = 0.45$; detection efficiency of SNSPDs at degenerate SPDC wavelength and averaged over different polarizations $\eta_{\text{detec}} \approx 0.6$; spectral detection factor for measurement with long-pass filter 1500 nm, $\eta_{\text{LP}} = 0.5$. The spectral detection factor takes into account that effectively half of the SPDC spectrum is detected when the long-pass filter at 1500 nm is inserted (compared to the spectrum in Fig. 4b).

## Fiber spectroscopy

Fiber spectroscopy was carried out to measure the photon-pair spectrum by mapping the spectral information onto the temporal domain using a dispersive medium. In this work, the dispersive medium consisted of two spools of SMF28 fiber (Corning), each with a length of 1 km.

The fiber spectroscopy experiment was conducted in two distinct configurations. In the first scenario, as shown in Fig. 2a, the photon pairs traveled through the same fiber spool. Following this, they were split using a 50:50 fiber beamsplitter before being detected by SNSPDs (Single Quantum Eos with timing jitter ≤25 ps). The arrival time differences were measured by a correlation electronics (qutools quTAG with timing jitter ≤10 ps). In the second configuration, as shown in Fig. 4a, the photon pairs were initially separated via a dichroic mirror. Subsequently, a 1 km dispersive fiber spool was introduced into each of the photon pathways, before detection through the SNSPDs. The group-velocity dispersion of the fiber leads to a time delay between signal and idler photons, which can be mapped to their wavelength difference[39]. Note that in this correlation-based measurement, always both photons of a pair need to be detected. An edge-pass filter, therefore, determines via energy conservation the entire width of the detected photon-pair spectrum. For instance, the spectrum measured in Fig. 2d using a long-pass filter with cut-on wavelength $\lambda_{\text{c}} = 1500$ nm and pump wavelength 788 nm fixes the long-wavelength edge of the spectrum to $(1/\lambda_{\text{p}} - 1/\lambda_{\text{c}})^{-1} = 1660$ nm.

## Quantum-state tomography

For the tomographic measurement of the two-photon polarization quantum state, both photons have to be projected into mutually unbiased bases. For this, we first separated signal and idler photons based on their frequency in our Hanbury Brown-Twiss interferometer (see Fig. 4a). A more detailed schematic of the experimental setup is provided in Supplementary Fig. 7. We implemented a dichroic mirror by using the reflection of a slightly tilted short-pass interference filter with cut-off wavelength 1600 nm (Edmund Optics #84-656). In each collection arm of the correlation setup, an arbitrary polarization basis could be set using a sequence of the achromatic quarter-wave plate (Thorlabs AQWP05M-1600), half-wave plate (Thorlabs AHWP05M-1600), and linear polarizer (Thorlabs WP25M-UB). During all changes in the polarization basis, the orientation of the linear polarizer was kept constant in order to avoid effects from the polarization sensitivity

of the detectors. For a full reconstruction, the state has to be measured in 16 different basis configurations. Please refer to Supplementary Note 5C for details on the chosen projection bases. We evaluated the measurements using a maximum likelihood method[42,43]. The uncertainty of the state concurrence $C$, derived from the experimentally measured density matrix, was determined using a Monte Carlo approach[56].

## Green's function method for pair-generation in layered materials

Our theoretical formalism is based on the Green's function (GF) quantization approach for the description of pair generation[57], where the coincidence detection probability at different spatial coordinates for a signal and idler photon generated by a nonlinear source through SPDC takes the form:

$$p_{GF}(\mathbf{r}_s, \mathbf{r}_i) \propto \left| \sum_{\alpha, \beta, \gamma} \sum_{\sigma_s, \sigma_i} d^*_{s, \sigma_s} d^*_{i, \sigma_i} \times \int d\mathbf{r} \chi^{(2)}_{\alpha\beta\gamma}(\mathbf{r}) \times E_{p, \gamma}(\mathbf{r}, \omega_s + \omega_i) G_{\sigma_s \alpha}(\mathbf{r}_s, \mathbf{r}, \omega_s) G_{\sigma_i \beta}(\mathbf{r}_i, \mathbf{r}, \omega_i) \right|^2,$$

(2)

where $\alpha$, $\beta$, and $\gamma$ indices run over the $x, y$, and $z$ directions. Here, $d_\sigma$ are the components of detection vector $\mathbf{d}$, where $\sigma = x, y, z$. $E_{p, \gamma}(\mathbf{r})$ are the vector components of the complex-valued monochromatic pump with frequency $\omega_p = \omega_s + \omega_i$. $G_{ij}(\mathbf{r}, \mathbf{r}', \omega)$ are the tensor components of the electric GF. Finally, $\chi^{(2)}_{\alpha\beta\gamma}(\mathbf{r})$ are the components of the second-order nonlinear tensor.

Here, the GF describes all the linear properties of the system and is incorporated into the quantum formalism to include nonlinear processes that involve the generation of entangled photons, such as SPDC. Due to the generality of the GF method, this formalism can describe any thickness of the 3R-MoS$_2$ nonlinear crystal, ultra-thin or thick, and it can be used to describe near- and far-field radiation in the non-paraxial regime[44,58]. Remarkably, this formalism allows us to keep track of any polarization and directionality effects in the pair-generation process, which makes it useful in the reconstruction of polarization states of entangled photons[45]. For modeling the 3R-MoS$_2$ crystal, we use the refractive index data provided in ref. 13 and the relative magnitude of the nonlinear tensor elements $d_{16}$ and $d_{31}$ from ref. 14. For a detailed discussion of the influence of the different tensor elements on the generated quantum states, refer to Supplementary Note 2.

## Data availability

Raw data that supports this study is available in the Supplementary Information and has also been deposited in the figshare database under accession code https://doi.org/10.6084/m9.figshare.26756398.

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

## Acknowledgements

This work was funded by the Deutsche Forschungsgemeinschaft (DFG, German Research Foundation) through the International Research Training Group (IRTG) 2675 "Meta-ACTIVE", project number 437527638 (T.P., F.S., F.E., I.S.), the Collaborative Research Center (CRC) 1375 "NOA" (T.P., F.S., F.E., I.S.), through "MEGAPHONE" project number 505897284 (S.Sar.) and through the Emmy Noether Program, project number STA 1426/2-1 (I.S.). The authors further acknowledge funding from the Bundesministerium für Bildung und Forschung (BMBF, German Federal Ministry of Education and Research) under the project identifiers 13N14877 (F.S.), 13XP5053A (F.E.), 16KISQ89 (F.E.); and by the State of Thuringia (Quantum Hub Thüringen, 2021 FGI 0043, T.P., F.S., F.E., I.S.). Furthermore, the authors acknowledge funding support from ANU PhD student scholarship (Y.T., H.Q.), Australian Research Council grant no. DP220102219 (Y.L.), LE200100032 (Y.L.), and ARC Centre of Excellence in Quantum Computation and Communication Technology (project number CE170100012, Y.L.). Additionally, this project has received funding from the European Union's Horizon 2020 research and innovation program under the H2020-FETOPEN-2018-2020 grant agreement no. [899673] (Metafast, T.P., I.S.).

## Author contributions

M.A.W. conceived the ideas, coordinated the measurements and theoretical modeling, designed the SPDC and tomography experiments, did the analytical calculations, analyzed the data, created the figures, and wrote the first draft of the manuscript under the supervision of F.E., Y.L., T.P, I.S., S.Sar. and F.S. Y.T. fabricated all samples. M.A.W., A.F., and S.Shi. did the polarization-resolved SPDC measurements. B.L., F.A., and A.F. did the polarized SHG experiment. M.A.W. did the tomography experiments. E.S. did the GF calculations under the supervision of S.Sar. A.F. measured the SHG map and S.R. measured the PL spectra. M.A., I.P.P, A.F., and M.A.W designed and calibrated the correlation fiber spectrometer. Y.T., H.Q., and J.J. designed and carried out the experiment for thickness-dependent SHG efficiency. B.L. did analytical calculations and contributed to writing the first manuscript draft. F.A. and S.Shr. did preliminary SHG and PL analysis for 3R-MoS$_2$ samples. F.E., Y.L., T.P, I.S., S.Sar., and F.S. acquired funding and provided experimental resources. F.E., F.S., S.Sar., Y.L., and A.F. provided major revisions to the manuscript. All authors discussed the results and contributed to the manuscript.

## Funding

## Competing interests

M.A.W., S.Sar., F.S., T.P., and F.E. are inventors on a patent application by the Fraunhofer Institute for Applied Optics and Precision Engineering IOF (63287DE) expanding on the results presented in this manuscript. The remaining authors declare no competing interests.

## Additional information

**Peer review information** : *Nature Communications* thanks Christiano de Matos and the other, anonymous, reviewer(s) for their contribution to the peer review of this work. A peer review file is available.

