## [Peer Review File · Nature Communications]

A Tunable Transition Metal Dichalcogenide Entangled Photon-Pair SourceEditorial note: Parts of this Peer Review File have been redacted as indicated to remove third-party material where no permission to publish could be obtained.

REVIEWER COMMENTS

Reviewer #1 (Remarks to the Author):

Weissflog et al. report the generation of entangled photon pairs using a bulk 3R-MoS₂ flake (thickness ~300 nm), a van der Waals crystal with high nonlinearity, and compact footprints. SPDC from thin films is not new, it has been reported in the last years in a few pioneering works.

However, different from previous report of SPDC from van der Waals compounds, Weissflog et al. directly show evidence of polarization entanglement measuring quantum state tomography. Beautiful!

The broadband generated maximally polarization entangled Bell states show a fidelity up to 96%.

The paper is well-written, the data and the derivation are clear, and the conclusions are adequately supported by the experimental observations.

I am in principle positive to recommend publication in Nature communications, however, there are some critical points that require clarification and revision.

Please see below.

Minor Revisions

1. In the introduction: the sentence “show a strong second-order nonlinear response” should be complemented with the actual value of the nonlinearity and related references. How strong is it? How does it compare to standard crystals used for SPDC?
2. Always in the introduction, please remove the sentence “To the best of our knowledge, this is the first realization of...”, the Nature publishing group does not accept expressions such as “the first”, “paves the way”, etc
3. ...and in the conclusions “In this work we observe for the first time SPDC in a transition metal dichalcogenide.” see comment 2.
4. Please mark the z-axis in Fig 1a,b.

5. The unit μm in Fig. 1d-e should not be formatted italic.

Revisions

1. Given the microscopic lateral size of the flake - how do the authors image the sample in the SPDC measurements to make sure the laser is actually exciting the region of interest marked in Fig. 1? From the illustration in Fig. 2a I cant see any imaging system coupled to the setup, neither I see it in the description in the main text. Do you use the same scanning system (1 μm step-width on a motorized XYZ-stage) of the SHG measurements? Even then, I would not see how practically it works when it comes to find a 100 μm x 100 μm sample on a 1 cm x 1 cm substrate. Please clarify.
2. What is the spotsize of the laser on the sample? Although SPDC mainly depends on the average pump power, the power itself is not necessarily as informative as the intensity, especially when it comes to compare SPDC and PL.
3. Following up - what's the pump intensity used to measure the PL spectra in Fig. 1b?
4. Polarization resolved measurements: "For SPDC detection without a polarizer, we observe the expected constant coincidence rate, independent of the pump polarization (Fig. 3(b))." perhaps here you should cite Ref. [18], as the exact same polarization dependence has been shown for OPA/DFG (i.e., seeded down-conversion) in monolayer TMDs.
5. Polarization resolved measurements: "Furthermore, by varying the pump polarization for several constant analyzer positions we obtain a two-lobed pattern" Can you expand on the implications and the significance of such a polarization dependence? Additionally, what happens if you keep the pump fixed and rotate the collection analyzer? Can the authors also provide this measurement for 2-3 different pump orientations with respect to the AC direction of the sample?
6. Polarization resolved measurements: in Fig. 3a some SHG data point look pretty off compared to the fit, in principle this measurement should be easy to do also with more data points.

Major Revisions

1. Fig S1d, the theory doesn't actually match with Ref. [5], could the authors clarify why? From the black "theory" curve shown in Fig. S1d the coherence length at 1550 nm seems to be ~ 800 nm, while it has been shown that is around 550 nm. Please clarify.

2. Along the previous comment, how did the authors choose the thickness of the flake in the present work? And why not choosing a flake that has a thickness close to the “constructive” peaks in the Fabry-Perot resonances?

3. In Fig. 4b the authors show the idler/signal spectrum, transmitted/reflected through/by a dichroic mirror with cutoff around 1550 nm. Why are these spectra so different than the one shown in Fig. 2d? My understanding is that in Fig. 2d signal and idler spectra are not separated, however the spectrum does not extend beyond 1680nm, while in Fig. 4b it seems that the idler spectrum extends to 1900nm.

With a 788nm pump the longest wavelength you can detect coincidences with is ~1660nm due to energy conservation. Can the authors motivate this choice?

4. One of my major concerns is the value of the CAR. It is extremely low, making this system practically useless. Could the authors compare their CAR to typical CAR values of standard SPDC sources like e.g. BBO, PPLN, PPKTP? What are the minimum requirements in real-world applications? How do you envision the use of TMDs in the future given such low performances? Complementing the manuscript with a short discussion about the state of the art of SPDC generation, and a comparison with this work, would make the paper stronger, and more accessible also to readers who are not necessarily working in the field of 2D materials.

5. The CAR values in S4.1 do not really follow the expected dependence $CAR=1/P_{\text{Pump}} + 1$. Why? The behavior looks linear to me.

In short, I believe this work is novel and impactful, and I believe it is of broad interest in the field.

I'd be happy to support publication if the authors implement the revisions above.

Reviewer #2 (Remarks to the Author):

The manuscript entitled “A Tunable Transition Metal Dichalcogenide Entangled Photon-Pair Source,” by Weissflog and co-workers report on the experimental demonstration of entangled photon pair generation via spontaneous parametric down conversion (SPDC) in a 285-nm-thick flake of MoS₂ in the 3R phase (which exhibits no inversion symmetry and, thus, second order nonlinearity). The authors first theoretically show that the symmetry of

the crystal (point group C_{3v}) straightforwardly allows for the generation of maximally entangled photons in two Bell states, as well as for polarization tunability without compromising the entanglement. Then, they experimentally confirm entanglement and polarization tunability, as well as observe the two predicted Bell states. In my opinion, the work is novel, timely and can be of interest to the wide scientific community working in the fields of quantum technologies, nonlinear optics and layered (2D) materials. The work seems to have been carried out with proper care, and the manuscript is clear and accurate. I would recommend its publication in Nature Communications after the authors address the following major issues:

1. Even though it is indeed interesting that Bell states can be directly obtained by simple pump polarization control, this is a property of any crystal in the D_{3h} (or, within reasonable approximation, the C_{3v}) point group. Hasn't this been previously explored even for bulk crystals with the same symmetry? If it has, please add references.
2. Along the same subject, the authors show that 2 out of the 4 states of the Bell basis can be directly obtained. They should comment on how the other 2 states can be obtained without significantly increasing the complexity of the system.
3. The spectrum in Figure 4b is significantly broader than that in Fig. 2d. Why is that? I assume that in the short wavelength edge the reason might be that the 1500nm cut-on filter is not being used (although this must be explicitly stated somewhere). But why does the long wavelength edge also stretch further in this figure?
4. In the conclusions' section, the authors state: "we show that for any linear pump polarization, a different, maximally entangled state is generated while the generation efficiency is independent of the pump polarization." They must clearly say that this is a theoretical result, as they only show experimental results with the pump polarized along x and y.
5. Also in the conclusions, the authors say that a possible way to increase the source brightness would be to exploit exciton resonances. But couldn't this reduce quantum

coherence? Please comment.

6. In Methods – Sample Fabrication: a reference must be given to the crystal synthesis method, and the substrate pre-treatment must be described.

7. In Methods - Polarization-Resolved SHG Measurements and SHG Mapping: the authors state that “Polarization-resolved SHG measurements were carried out with the same setup as used for quantum measurements but working in reverse: the fundamental beam was incident from one of the collecting fibers and[...]” Does this mean that the femtosecond pulse passes through 1 km of single mode fiber? This would significantly alter its temporal and spectral characteristics. I believe this is not the case. Please clarify.

8. In Supplementary Note S2, the authors say that the $\chi(2)$ dispersion in the spectral range of interest is negligible; but later they attribute differences in the $\chi(2)$ values measured in references [9] and [10] to a high $\chi(2)$ dispersion. How are these two statements compatible? Even if the $\chi(2)$ dispersion is only significant in the ranges measured in the references, how do the authors know that the d_{16}/d_{31} ratio measured in ref. [9] is maintained at other frequencies ranges, despite dispersion? This is an important issue and must be carefully addressed.

Some other (minor) points:

a) It would be good to mention that concurrence and fidelity are defined in the Supplementary Information right after these parameters are first mentioned in the main text. This information is currently given a bit too late.

b) Just as a reference to readers, I suggest that the authors mention the coherence length between pump, signal and idler in the main text.

c) The green area in Fig. 2b should be explained in the figure caption.

d) In the second paragraph of section I-C, the parameter t (in “ $t=285$ nm”) appears without

ever being defined. I suppose that this is the crystal thickness, but this should be explicitly defined.

Reviewer #3 (Remarks to the Author):

In the manuscript by M.A. Weissflog et al., the authors demonstrate the generation of polarization-entangled photon pairs through a submicron transition metal dichalcogenide (TDM) crystal. The work presented in this manuscript pioneers the demonstration of spontaneous parametric down conversion (SPDC) process in TDM crystals, while previous literature has been limited to second harmonic generation (SHG). The authors claim and experimentally show the realization of maximally entangled Bell states, specifically $|\Phi^{--}\rangle$ and $|\Psi^{++}\rangle$, which can be respectively obtained by pumping the crystal with a y-polarization and x-polarization pump. By pumping the crystal with a ϕ_p -rotated pump polarization, the authors show that a maximally entangled state in a superposition of $|\Phi^{--}\rangle$ and $|\Psi^{++}\rangle$ can be generated, as well, with same fidelity. The demonstration of maximal entanglement is proved through concurrence estimation (described in all its details in the Supplementary Information), theoretically for all states ($|\Phi^{--}\rangle$, $|\Psi^{++}\rangle$, and their superposition) and experimentally for the $|\Phi^{--}\rangle$ and $|\Psi^{++}\rangle$ states. The authors further measure quantum state tomography and fidelity to validate the generation of states $|\Phi^{--}\rangle$ and $|\Psi^{++}\rangle$. Coincidence-to-accidental ratio is estimated to validate the quantum sources at different pump powers, while the Green's function quantization approach is theoretically utilized to describe photon pair generation.

This work can have a considerable impact in quantum photonics, especially with respect to the development of integrable entangled photon sources utilizing SPDC. In turn, the reported results can bring contribution to quantum technologies relying on highly efficient, bright, integrated sources of entangled photons. Considering these aspects and the importance of the work for specialized fields, I recommend the manuscript by M.A. Weissflog et al. for publication to Nature Communications.

Some relatively minor comments and questions should be considered before submission,

which are listed in the following.

- The entangled state in Eq. (1) of the main text is a superposition of two Bell states. It is a maximally entangled state while it does not belong to multipartite entangled states (e.g., genuine multipartite or hyperentangled states). Is there a targeted applications the authors have in mind for such a state? Is it more convenient to select precise angles ϕ_p or set them to 0° or 90° to obtain the standard Bell states?

- With respect to this, the quantum state in Eq. (1) is only reported theoretically. Is there any experimental limitation for not demonstrating a state with a value of ϕ_p which is not either 0° or 90° and evaluating quantum state tomography and concurrence (measured experimentally)?

- On line 116 of the main text, the authors write that they “choose an area far away from the crystal edges and all cracks”. Was there a quantitative parameter/criterion for choosing such an area?

- In line 135, the authors write that a “ $CAR > 2$ together with the linear scaling of the coincidence rate of the coincidence rate with the pump power is clear evidence for the SPDC origin of the coincidence peak”. Why exactly 2? Is this something related to the source or the pumping scheme?

- In the CAR plot in Fig. S3 of the Supplementary Information, the car value at 5 mW seems a bit large compared to other pump powers, unless I am missing something.

Letter of Response for Manuscript
“A Tunable Transition Metal Dichalcogenide Entangled Photon-Pair Source” (NCOMMS-24-02555-T)

We would like to thank the reviewers for their time and the careful reading of our manuscript and the generally positive response. We particularly would like to let the reviewers know that we perceived their reports and suggestions as very constructive and fair and believe that they improved the manuscript.

Below, we reply to the reviewers’ comments and questions (listed in **blue**). Our replies are given in **black** font. In the revised manuscript, the changes addressing the reviewers’ comments are in **red** font.

On behalf of the authors,
Maximilian Weissflog

Reviewer #1 (Remarks to the Author):

Weissflog et al. report the generation of entangled photon pairs using a bulk 3R-MoS₂ flake (thickness ~300 nm), a van der Waals crystal with high nonlinearity, and compact footprints. SPDC from thin films is not new, it has been reported in the last years in a few pioneering works. However, different from previous report of SPDC from van der Waals compounds, Weissflog et al. directly show evidence of polarization entanglement measuring quantum state tomography. Beautiful! The broadband generated maximally polarization entangled Bell states show a fidelity up to 96%. The paper is well-written, the data and the derivation are clear, and the conclusions are adequately supported by the experimental observations. I am in principle positive to recommend publication in Nature communications, however, there are some critical points that require clarification and revision. Please see below.

Minor Revisions 1. In the introduction: the sentence “show a strong second-order nonlinear response” should be complemented with the actual value of the nonlinearity and related references. How strong is it? How does it compare to standard crystals used for SPDC?

The point raised by the reviewer is indeed important and we will address it carefully. The reason that we didn't previously include a comparison of $\chi^{(2)}$ of 3R-MoS₂ with other materials into the manuscript was that such comparisons have been done in several previous works (compare e.g. Q. Guo et al., ‘Ultrathin quantum light source with van der Waals NbOCl₂ crystal’, Nature, 613, 7942, (2023), Ref. 25). Furthermore, several factors pose challenges when obtaining and comparing absolute values for the $\chi^{(2)}$ of TMDs in general, leading to a considerable spread of reported nonlinear coefficients in the literature. Since some care must be taken for properly citing and interpreting and comparing these values, we want to briefly mention some important factors.

Firstly, $\chi^{(2)}$ in TMDs is highly dispersive due to the excitonic resonances or higher-energy transitions in the band-nesting region, which makes the comparison of nonlinear coefficients of TMDs with other materials highly dependent on the wavelength. We will also further comment on this in the answer to reviewer #2, qu. 8. Secondly, the very high-refractive index of the material leads to the formation of Fabry-Perot cavities, which makes nonlinear measurements very sensitive to flake thickness and wavelength used (and the method how is this accounted for when extracting the nonlinear susceptibility).

A third parameter related to the sample thickness is absorption at the second-harmonic wavelength, since many measurements are carried out in the lossy region of 3R-MoS₂ (approx. for $\lambda < 740$ nm). Since this region offers excitonic enhancement, this can lead to very high values of $\chi^{(2)}$ for TMDs when compared with many other materials. This however is only a practical advantage when working with ultra-thin devices with thicknesses below the absorption (penetration) depth of the material. For comparison with transparent materials which can be used for nonlinear devices with long propagation length this would be an ‘unfairly’ high value to compare to (a fact we feel is not always emphasized adequately in the literature).

After giving this perspective, we summarize in Table 1 the largest values of $\chi^{(2)}$ or the nonlinear conversion efficiency density η , respectively, reported for 3R-MoS₂. The conversion efficiency density is defined as $\eta = \frac{P_{SH}}{P_{FW} \times L^2}$ (X. Xu et al., Nat. Photon., 16, 10, (2022), Ref. 13), i.e. is the second-harmonic power P_{SH} normalized to the fundamental wave power P_{FW} and propagation length L .

Table 1: Literature overview second-order nonlinear susceptibility 3R-MoS₂

Fundamental wavelength (nm)	$\chi^{(2)}$ (pm/V)	η (% W ⁻¹ cm ²)	Sample thickness	SHG in transparency region	Reference
1521	-	71800	4.2 nm	Yes	[1] X. Xu et al., 'Towards compact phase-matched and waveguided nonlinear optics in atomically layered semiconductors', Nat. Photon. , 16 , 10, (2022) (Ref. 13)
1450	100 pm/V	-	293 nm (multi-flake stack with quasi-phasematching)	Yes (close to absorption edge)	[2] C. Trovatiello et al., 'Quasi-phase-matched up- and down-conversion in periodically poled layered semiconductors'. arXiv, Dec. 31, 2023. doi: 10.48550/arXiv.2312.05444. (preprint published after initial posting of our work, newly added reference)
1200	30	-	124 nm	No	[3] J. Shi et al., '3R MoS ₂ with Broken Inversion Symmetry: A Promising Ultrathin Nonlinear Optical Device', Advanced Materials , 29 , 30,1701486, (2017)
1064	1000±400	-	10 μm	No	[4] G. A. Wagoner, P. D. Persans, E. A. Van Wagenen, and G. M. Korenowski, 'Second-harmonic generation in molybdenum disulfide', J. Opt. Soc. Am. B , 15 , 3, p. 1017, (1998) (Ref. 14)
950	850	-	9 nm	No	[5] G. Zograf, A. Y. Polyakov, M. Bancerek, T. Antosiewicz, B. Kucukoz, and T. Shegai, 'Combining ultrahigh index with exceptional nonlinearity in resonant transition metal dichalcogenide nanodisks'. arXiv, Aug. 22, 2023. doi: 10.48550/arXiv.2308.11504. (newly added reference)
800	1000	-	70 nm, 170 nm	No	[6] E. Mishina et al., 'Observation of two polytypes of MoS ₂ ultrathin layers studied by second harmonic generation microscopy and photoluminescence', Applied Physics Letters , 106 , 13, p. 131901, (2015)

This table reflects the previously discussed complication with the large spread of reported values for the $\chi^{(2)}$ of 3R-MoS₂. However, the trend of the nonlinear susceptibility fits to the SHG measurement with fundamental wavelengths varying from 800 nm to 1600 nm measured by [1] X. Xu et. al. (Fig. 2d in their paper <https://www.nature.com/articles/s41566-022-01053-4>). Note however, that their measurement is not calibrated for absolute values of the nonlinear susceptibility.

In Table 2, we compare the values for 3R-MoS₂ with the nonlinear coefficient for nonlinear materials commonly used in nonlinear (quantum) light sources. In particular, we list beta barium borate (beta-BaB₂Bo₄, BBO), potassium titanyl phosphate (KTP), lithium niobate (LN), gallium arsenide (GaAs) and gallium phosphide (GaP).

Table 2: Literature overview second-order nonlinear susceptibility commonly used nonlinear materials

Material	Fundamental wavelength (nm)	chi ² (pm/V)	sample thickness	SHG in transparency region	Reference
BBO	1064	3.88 (effective)	bulk	yes	R. C. Eckardt, et. al., , IEEE Journal of Quantum Electronics, 26 , 5, pp. 922–933, (1990) (newly added reference)
KTP	1064	29.2	bulk	yes	I. Shoji, T. Kondo, A. Kitamoto, M. Shirane, and R. Ito, 'Absolute scale of second-order nonlinear-optical coefficients', Journal of the Optical Society of America B, vol. 14, no. 9, p. 2268, Sep. 1997, doi: 10.1364/JOSAB.14.002268. (newly added reference)
	1313	22.2	bulk	yes	
LN	1064	49.8	bulk	yes	
	1313	40.6		yes	
GaAs	1064	340	bulk	no	
	1533	238	bulk	no (close to absorption edge)	
GaP	1064	141.2	bulk	yes	
	1313	73.6	bulk	yes	

We hope that this discussion provides a sufficient background to put the nonlinear properties of 3R-MoS₂ into correct perspective with respect to other materials. The statement in line 68+69 the reviewer mentioned (TMDs “show a strong second-order nonlinear response”) actually referred to TMDs in the monolayer limit, where due to several reasons quantifying $\chi^{(2)}$ is even more challenging, leading to large discrepancies in the literature (see e.g. A. Säynätjoki et al., Nat. Commun., **8**, 1, p. 893, (2017), in section “SHG and THG characterization” for a discussion). We therefore reformulate this statement and do not mention the nonlinearity strength of monolayer TMDs anymore.

In the monolayer limit, TMDs with the structural form MX₂ (M=Mo,W, X=S,Se) are non-centrosymmetric and have a crystal lattice with three-fold rotational symmetry around the z-axis, corresponding to the point group D_{3h}.

Instead, we insert a statement about the absolute magnitude of the nonlinearity of 3R-MoS₂ into the introduction. In our view, the important point of our current work is the observation of photon-pairs and the characterization of their entanglement properties and not primarily quantifying and discussing $\chi^{(2)}$, which was the focus of many of the cited works. We therefore summarize this discussion for the main manuscript as briefly as possible:

The focus of this work is the demonstration of fundamental properties of generating entangled quantum states in 3R-MoS₂ and not yet a highly efficient and integrated device design. However, in general the material is well suited for designing very compact, highly efficient photon-pair sources.

The high second-order nonlinearity of 3R-MoS₂ ($\chi^{(2)} \approx 100 \text{ pm/V}$ [26] in transparency region and peak values $\chi^{(2)} > 800 \text{ pm/V}$ [14,26,27] with excitonic enhancement in absorbing region) is at the

same level or largely exceeds the nonlinearity of many established materials (beta barium borate (BBO) $\chi^{(2)} = 3.9 \text{ pm/V}$ [28], potassium titanyl phosphate (KTP) $\chi^{(2)} = 29.2 \text{ pm/V}$, lithium niobate (LiNbO₃) $\chi^{(2)} = 49.8 \text{ pm/V}$, gallium arsenide (GaAs) $\chi^{(2)} \approx 340 \text{ pm/V}$ (absorbing) and gallium phosphide (GaP) $\chi^{(2)} \approx 141 \text{ pm/V}$ all at fundamental wavelength 1064 nm} ref. [29]). Leveraging this, it was demonstrated with second-harmonic generation (SHG) that 3R-MoS₂ requires a two orders of magnitude shorter propagation length to reach the same second-order nonlinear conversion efficiency in the telecom range as LiNbO₃ [13]. While in this case crystals of only one coherence length were compared, the conversion efficiency of 3R-MoS₂ photon-pair sources can be scaled to the required level e.g. through quasi-phasematching. Similar to periodic poling in ferroelectric materials, the nonlinearity in stacks of several multilayer 3R-MoS₂ crystals can be periodically modulated by suitably rotating consecutive crystals [19], which was experimentally demonstrated very recently [26].

For this discussion, we added the following new references to the main paper.

- C. Trovatiello et al., ‘Quasi-phase-matched up- and down-conversion in periodically poled layered semiconductors’. arXiv, Dec. 31, 2023. doi: 10.48550/arXiv.2312.05444. (preprint published after initial posting of our work)
- G. Zograf, A. Y. Polyakov, M. Bancerek, T. Antosiewicz, B. Kucukoz, and T. Shegai, ‘Combining ultrahigh index with exceptional nonlinearity in resonant transition metal dichalcogenide nanodisks’. arXiv, Aug. 22, 2023. doi: 10.48550/arXiv.2308.11504.
- R. C. Eckardt, et. al., , IEEE Journal of Quantum Electronics, **26**, 5, pp. 922–933, (1990)
- Shoji, T. Kondo, A. Kitamoto, M. Shirane, and R. Ito, ‘Absolute scale of second-order nonlinear-optical coefficients’, Journal of the Optical Society of America B, vol. 14, no. 9, p. 2268, Sep. 1997, doi: 10.1364/JOSAB.14.002268.

2. Always in the introduction, please remove the sentence “To the best of our knowledge, this is the first realization of...”, the Nature publishing group does not accept expressions such as “the first”, “paves the way”, etc

We thank the reviewer for pointing us towards that policy, which we weren’t aware of. We removed that passage accordingly.

3. ...and in the conclusions “In this work we observe for the first time SPDC in a transition metal dichalcogenide.” see comment 2.

We reformulated the sentence to:

In this work we observe **photon-pair generation via** SPDC in a transition metal dichalcogenide.

4. Please mark the z-axis in Fig 1a,b.

This is indeed a good idea to make our coordinate definition clearer. We marked it accordingly.

5. The unit μm in Fig. 1d-e should not be formatted italic.

We fixed the formatting.

Revisions

1. Given the microscopic lateral size of the flake - how do the authors image the sample in the SPDC measurements to make sure the laser is actually exciting the region of interest marked in Fig. 1? From the illustration in Fig. 2a I cant see any imaging system coupled to the setup, neither I see it in the description in the main text. Do you use the same scanning system (1 μm step-width on a motorized XYZ-stage) of the SHG measurements? Even then, I would not see how practically it works when it comes to find a 100 μm x 100 μm sample on a 1 cm x 1 cm substrate. Please clarify.

We thank the reviewer for pointing out that we indeed missed to describe this part. Of course they are right, in our experimental setup we have an imaging system to locate the TMD crystal on our sample. We did not include the sample imaging arm into the schematic of the optical setup in Fig. 2a and Fig. 4a since it is not really related to the physics and we were concerned this might overcomplicate the drawing. However, from a practical viewpoint this is of course a relevant question. In Figure 1 we show a detailed schematic of the experimental setup that also includes the imaging part.

[figure redacted]

Figure 1: Detailed schematic of the experimental setup used for measurements with a single collection fiber and a fiber beamsplitter. The sample can be imaged via a camera that can be inserted into the beampath using a 50:50 beamsplitter on a flip mount. In this case, the sample is illuminated via a white light source which is coupled into the beam-path via a second flip mirror. The beam-path of the white light illumination and image, respectively, is marked with grey arrows.

In the following we briefly outline the procedure for aligning the setup and finding the measurement position. As rough navigation aids, we have four large markers on the substrate, which bound the area where the TMD crystal is located, see Figure 2 a). After finding these large markers in the imaging system of the optical setup, we can find the TMD crystal within a reasonably short time. Figure 2 b) shows a screenshot of the TMD crystal as observed in the home-built imaging system of the setup. Although the imaging quality does not reach the level of a commercial microscope (see Fig. 1 d of the main manuscript), the image quality is sufficient to find a particular measurement position on the sample.

Figure 2: a) Photograph of the sample inserted into the sample holder used in the experiment. The position of the TMD crystal is marked by 4 lines on the substrate. b) TMD crystal as observed by the imaging arm of the optical setup.

Due to the low brightness of the SPDC source, the alignment of the setup is first carried out with classical laser beams at $\lambda = 1550$ nm (close to the degenerate SPDC wavelength) and without the sample. The alignment beams are first coupled from the pump fibre launcher into the detector fiber and vice versa, to ensure a good pre-alignment of pump and detector fiber mode. In this step all essential components (fiber coupling lenses, focusing and collection lens etc. are adjusted). However, since the SPDC pump beam has a wavelength of $\lambda = 788$ nm, the pre-alignment with $\lambda = 1550$ nm would not ensure an optimal lens position yet. Therefore, a femto-second pulsed laser with central wavelength $\lambda = 2 \times 788$ nm = 1576 nm is now launched into a short (0.5 m) fiber, which is placed in the detector fiber holder. After inserting the sample, we can use this to excite SHG at 788 nm in the TMD crystal, which we observe on the camera. The positioning of all lenses is optimized by observing the SHG intensity. The SHG spot marks the area ‘seen’ by the detector fiber mode, which allows us to directly see the area from which the single-photon detectors will collect. This enables navigation to a desired measurement position on the sample. In a last step, the flip-components for the imaging system are removed from the beam path and the SHG signal is coupled to the same fiber that is used for the SPDC pump laser.

Although on first sight this multi-step process of alignment with guide-lasers and then SHG measurements might look somewhat complicated, in our experience this is a reliable way to achieve single-mode fiber coupling of a weak, parametric photon-pair source. In particular, the intermediate step with doing SHG is helpful when aligning for SPDC detection with two separate detector fibers as done in our tomography measurement. Here, we excite SHG with pump pulses from both detector fibers simultaneously, and spatially overlap them using the camera. This ensures that both detector fibers collect from the same measurement position. For completeness, we also show the detailed schematic of the experimental setup for the quantum state tomography in Figure 3.

[figure redacted]

Figure 3: Detailed schematic of the experimental setup used for quantum state tomography with a dichroic mirror and two separate fiber collection points. The spatial overlap of both fiber detection modes at the desired measurement location is ensured by exciting SHG with femto-second pump pulses emitted from both detector fibers. The generated SHG emission can be observed on the camera which allows to optimize the overlap of both fiber modes.

In the manuscript, we add the following additional explanation in the methods section “Photon-Pair Correlation Measurements”.

A more detailed schematic of the experimental setup is provided in supplementary Fig. S2. Excitation photons from a continuous-wave laser at $\lambda_p = 788 \text{ nm}$ (diode laser, Thorlabs FPL785P) were sent through a linear polarizer and a half-wave plate for pump polarization control and focused onto the sample by an aspheric lens with numerical aperture NA=0.4 (Thorlabs C110TMD-B). The measurement position is imaged in the experimental setup in a separate imaging arm via the same camera (Excelitas pco.edge 4.2 bi) as also used for SHG measurements. Compare supplementary Fig. S2 for details.

We also add a similar statement to the methods section “Quantum-State Tomography”:

A more detailed schematic of the experimental setup is provided in supplementary Fig. S3.

In the supplementary file, we add an additional supplementary note “Supplementary Note 4: Details of Experimental Setup for Photon-Pair Measurements” which essentially contains Figure 1 (labelled as Fig. S2) and Figure 3 (labelled as Fig. S3) as shown in this reply letter. Additionally, we also add the exact model number of all relevant optical components to the method section.

2. What is the spotsize of the laser on the sample? Although SPDC mainly depends on the average pump power, the power itself is not necessarily as informative as the intensity, especially when it comes to compare SPDC and PL.

The diffraction limited $1/e^2$ pump beam radius r_p on the sample is about $r_p \approx 1.3 \mu\text{m}$. The peak intensity of a Gaussian beam with this radius and power P follows from $I_p = \frac{2P}{\pi r_p^2}$. For a pump power of P=17.2 mW as e.g. used for the measurement reported in Fig. 2c, this corresponds to a peak intensity of $I_p = \frac{2 \times 0.0172 \text{ W}}{\pi \times (1.3 \times 10^{-4} \text{ cm})^2} \approx 648 \frac{\text{kW}}{\text{cm}^2}$. We mention now both values, pump power, and peak intensity in the manuscript.

... and a pump power incident on the sample of 17.2 mW (peak intensity ~648 kW cm⁻²).

3. Following up - what's the pump intensity used to measure the PL spectra in Fig. 1b?

In the experimental setup we use for PL measurements, we have a 1/e² beam spot diameter of 1 μm and a pump power at the sample of 26.2 mW. With this, the pump intensity is $= \frac{2 \times 0.02618 \text{ W}}{\pi \times (1 \times 10^{-4} \text{ cm})^2} \approx 1.67 \frac{\text{MW}}{\text{cm}^2}$. We noted the pump intensity in the main text when describing the PL measurement:

We observe no photoluminescence signal distinguishable from the detector darkcounts beyond 1.3 μm (see green shaded area Fig.2(b), pump intensity 1.67 MWcm⁻²)

4. Polarization resolved measurements: “For SPDC detection without a polarizer, we observe the expected constant coincidence rate, independent of the pump polarization (Fig. 3(b)).” perhaps here you should cite Ref. [18], as the exact same polarization dependence has been shown for OPA/DFG (i.e., seeded down-conversion) in monolayer TMDs.

This is a good point, we have added the reference in this section accordingly.

For SPDC detection without a polarizer, we observe the expected constant coincidence rate, independent of the pump polarization (Fig. 3(b))[18].

5. Polarization resolved measurements: “Furthermore, by varying the pump polarization for several constant analyzer positions we obtain a two-lobed pattern” Can you expand on the implications and the significance of such a polarization dependence? Additionally, what happens if you keep the pump fixed and rotate the collection analyzer? Can the authors also provide this measurement for 2-3 different pump orientations with respect to the AC direction of the sample?

The points raised by the reviewer here are indeed interesting, and we admit that we didn't explain this part in much detail in the manuscript due to the restricted space. However, the reviewer question motivates us to expand now more on this.

Before going into details of the raised question, let's recall that all polarization properties, particularly the generation of polarization entangled quantum states, are governed by the symmetry of the nonlinear tensor of 3R-MoS₂. Ultimately, in this work we are interested in quantum entanglement which requires state tomography for its characterization. However, the simpler polarization analysis with a single polarizer as reported in Fig. 3 of the main manuscript is still instructive. It helps relating the SPDC process to polarization characteristics well-known from SHG measurements and simultaneously already bears first hallmarks of the possibility of generating entangled states, as we will discuss in the following.

Generally, the (normalized) SPDC rate R_{SPDC} when measuring both signal and idler through the same, linear polarizer follows the relation

$$R_{\text{SPDC}} = \frac{1}{2} \sin^2(2\varphi_{\text{pol}} + \varphi_p). \quad (1)$$

Here, φ_{pol} marks the rotation angle of the linear polarizer that signal and idler are sent through, while φ_p is the polarization angle of the pump photon. The derivation of equation (1) has been given in detail in supplementary section S1 and will therefore not be repeated here.

The SPDC rate is periodic in φ_{pol} with period $\frac{\pi}{2}$ and periodic in φ_p with period π . As visualization we plot eq. (1) in Figure 4. We highlight again that this formula is intrinsically linked to the nonlinear tensor of the considered TMD. Confirming the validity of this dependence of the SPDC rate in eq. (1) experimentally would therefore be a first step towards showing that this nonlinear tensor allows to generate the polarization states we seek, without doing a full state tomography yet. We therefore performed several measurements that sample the two-variable function eq. (1) along specific “cut-lines” in the space spanned by $\{\varphi_{pol}, \varphi_p\}$. Let’s start with considering the case of rotating polarizer and pump polarization simultaneously as presented in Fig. 3c. The two cases we probed correspond to setting either $\varphi_{pol} = \varphi_p$ (red dashed line Figure 4, orange markers in Fig. 3c of main manuscript) or to setting $\varphi_{pol} = \varphi_p + \frac{\pi}{2}$ (grey dashed line Figure 4, blue markers in Fig. 3c of main manuscript). Both cases are diagonal cut lines in the space spanned by $\{\varphi_{pol}, \varphi_p\}$. Given the theoretically derived dependence, we expect a 3-lobed harmonic function in the interval $[0^\circ, 180^\circ]$ for both cases, which is in very good agreement with our measurement (see main manuscript).

Figure 4: Dependence of R_{SPDC} on the polarizer rotation angle φ_{pol} and pump polarization angle φ_p , respectively. Dashed coloured lines mark the configurations for which the measurements presented in Fig. 3 of the main manuscript have been taken. Here, either the analyzer is fixed at particular angles φ_{pol} (green dashed lines, Fig. 3d-f in main manuscript), or pump and analyzer polarization are co-rotating with $\varphi_{pol} = \varphi_p$ (red dashed line) or $\varphi_{pol} = \varphi_p + \frac{\pi}{2}$ (grey dashed line), respectively (Fig. 3c in main manuscript).

In principle, this sampling along different diagonal cut lines would already be a strong experimental validation for the derived dependence eq. 1. As further confirmation, we also probed a different case, which is fixed angles φ_{pol} and a rotating pump (green dashed lines in Figure 4, Fig. 3d-f of main manuscript). This corresponds to vertical cut lines, see Figure 4. Based on this, we expect to find a two-lobed function with π -periodicity, which is again in very good agreement with the experimental outcome. The reviewer raised now the very valid question what happens in the third possible configuration, which would be a constant pump polarization and a rotating analyser angle φ_{pol} , which we indeed didn’t comment on in the manuscript. However, by continuing the previous line of reasoning, we believe that the outcome of such a measurement can be predicted straightforwardly. For a constant pump polarization, we would sample eq. (1) along horizontal lines, and therefore find a $\frac{\pi}{2}$ -periodic dependence. For visualizing this, we plot the resulting functions in Figure 5 for fixed pump angles of $\varphi_p = 0^\circ$, $\varphi_p = 60^\circ$ and $\varphi_p = 120^\circ$, respectively.

For this particular relative orientation of polarizer and pump angle we did not do measurements so far. We agree with the reviewer that it would in principle be nice to also measure these dependencies in an experiment. Thinking this further, the best-case scenario would actually be to reproduce the entire functional dependence of eq. (1) in an experiment. However, based on the experimental results we have already presented we concluded that we can say with a high level of confidence that eq. (1) is experimentally confirmed. Since from there all possible relative orientations of φ_{pol} and φ_p can be predicted, we feel that further measurements of this type would not significantly expand the understanding of what to expect when doing SPDC measurements with a common linear polarizer. Furthermore, due to logistical reasons in our lab, doing such a measurement in addition to the new measurements we already took in response to reviewer #1, qu. 6, would significantly delay the revision of the manuscript right now. In sum we therefore decided to not add a new series of measurements for this point. We however fully agree with the reviewer, that the explanation in the text in this regard should be expanded.

Figure 5: Evolution of SPDC rate for pump polarization fixed to a) $\varphi_p = 0^\circ$, b) $\varphi_p = 60^\circ$, c) $\varphi_p = 120^\circ$ and rotating the analyzer.

Picking up the important points of the discussion above, we modify the explanation in the main manuscript in the following way.

For measurements through a common analyzer for both signal and idler, the symmetry of the nonlinear tensor leads to a dependence of the SPDC rate R_{SPDC} on pump polarization angle φ_p and analyzer angle φ_{pol} as $R_{SPDC} \propto \frac{1}{2} \sin^2(2\varphi_{pol} + \varphi_p)$ (see supplementary note S1 for the derivation). To experimentally verify this, we insert an analyzer in front of the fiber and simultaneously rotate pump and analyzer either in a parallel configuration $\varphi_{pol} = \varphi_p$ (orange squares in Fig. 3(c)) or perpendicular configuration $\varphi_{pol} = \varphi_p + \frac{\pi}{2}$ (blue dots in Fig. 3(c)). Both yield a characteristic six-fold, co-aligned pattern which matches the theoretically expected dependence of the form $R_{SPDC} \propto \frac{1}{2} \sin^2(3\varphi_{pol})$ when analyzed in terms of the SPDC pump polarization angle φ_p . Note that this is fully consistent with frequently reported SHG measurements that show a 30° shift between measurements with parallel or perpendicular polarizer angle [16], see supplementary note S1 for a more detailed discussion. Furthermore, by varying the pump polarization for several constant analyzer positions we obtain a two-lobed pattern (see Fig. 3(d-f)), further confirming the theoretically derived polarization dependence.

In supplementary note S1 we also discuss the polarization dependence when keeping the pump polarization angle constant and only rotating the analyzer. The raw coincidence histograms for all results in Fig. 3 are found in supplementary Figs.S4 and S5.

To not increase the length of the discussion in the main text too much, we include more details into supplementary note S1, where we add the following text and also Figure 4, Figure 5 and Figure 6.

This is a function periodic in φ_p with period π and periodic in φ_{pol} with period $\frac{\pi}{2}$. To visualize this, we plot eq. S14 in Fig. S1. The measurements reported in Fig. 3 of the main manuscript correspond to 'cut-lines' that sample the two-variable function for different configurations. The measurements with rotating pump and polarizer in parallel or perpendicular configuration (Fig. 3(c) of the main text) are marked with red and gray dashed lines, respectively. The configurations with fixed analyzer and rotating pump (Fig. 3(d)-(f) of the main text), are highlighted with green dashed lines in Fig. S1. The very good agreement between the predicted and experimentally measured functional dependence confirms eq. S14

Based on this, also the dependence for an experimental setting that we didn't explore, namely fixed pump polarization and rotating polarizer, can be evaluated. This would correspond to horizontal 'cut-lines' in Fig. S1. This leads to a characteristic four-lobed pattern of the SPDC rate, which we visualize in Fig. S2.

A last point related to the reviewer's question of the significance of the polarization dependencies, is how to interpret the co-alignment of the characteristic 6-lobed patterns for the SPDC measurements with co-rotating polarizer and pump. When discussing this work during conference presentations we noted more than once that this alignment of the polarization patterns initially led to confusion. Many listeners had apparently intuitively expected to see the same result as for SHG measurements, where the six-lobed patterns are usually rotated by 30° with respect to each other, when comparing configurations with parallel and perpendicular analyzer (compare e.g. Y. Li, et. al. Nano Lett. 13(7), 3329–3333 (2013) (Ref. 16 in main text, Ref. 4 in SI)). In fact, our results for polarized SPDC measurements are in full agreement with this (and have to due to quantum-classical correspondence). However, to see this, one has to consider that the role of the pump photon angle and the analyzer angle are swapped in the SPDC case as compared to SHG measurements. After discussions with peers we have the impression that this connection is not as immediately visible as we initially thought, which is why we inserted a brief explanation together with a supporting Figure 6 into the SI.

Lastly, we take a closer look at the importance of the reference angle in case of co-rotating pump polarization and analyzer angle. From eq. (S14) it is straightforward to see that for an alignment of the pump polarization either in parallel $R_{SPDC,\parallel}$ ($\varphi_{pol} = \varphi_p$) or in orthogonal configuration $R_{SPDC,\perp}$ ($\varphi_{pol} = \varphi_p \pm \pi/2$), the dependence is $R_{SPDC,\parallel} = \frac{1}{2} \sin^2(2\varphi_p + \varphi_p) = R_{SPDC,\perp} = \frac{1}{2} \sin^2\left(2\left(\varphi_p \pm \frac{\pi}{2}\right) + \varphi_p\right) = \frac{1}{2} \sin^2(3\varphi_p)$. This curve is shown in Fig. S3(a) and matches the experimental results as shown in Fig. 3(c) of the main text. We note here, that the characteristic six-fold pattern of the SPDC rate is co-aligned for both parallel and perpendicular analyzer configuration, when using the polarization angle of the SPDC pump photon φ_p as reference. This is in fact in full agreement with measurements of polarized SHG from TMDs, that reported a shift of 30° of the 6-lobed patterns obtained for parallel or perpendicular excitation [4]. To illustrate this, we also look at the case where the rotation angle of the analyzer φ_{pol} is used as reference. Then we find for the parallel configuration with $\varphi_p = \varphi_{pol}$ the dependence $R_{SPDC,\parallel} = \frac{1}{2} \sin^2(2\varphi_{pol} + \varphi_{pol}) = \frac{1}{2} \sin^2(3\varphi_{pol})$.

However, the perpendicular configuration where $\varphi_p = \varphi_{pol} \pm \frac{\pi}{2}$ yields $R_{SPDC,\perp} = \frac{1}{2} \sin^2 \left(2\varphi_{pol} + \varphi_{pol} \pm \frac{\pi}{2} \right) = \frac{1}{2} \cos^2(3\varphi_{pol})$. Therefore, when referenced to the analyzer angle φ_{pol} , the two polarization patterns have a shift of 30° , see the plot in Fig. S3(b). In our SPDC measurement the angle of the analyzer φ_{pol} selects the polarization direction of the down-converted (low-energy) signal and idler photons. In a typical SHG measurement, these low-energy photons however act as the pump. Therefore, when directly comparing SPDC measurements with SHG measurements plotted in terms of the SHG pump angle, see e.g. our measurement reported in Fig. 3(a) of the main text, a direct equivalence is obtained when referencing the SPDC measurement to the analyzer angle φ_{pol} .

Figure 6: Theoretical dependence of SPDC rate measured through a common polarizer for signal and idler photons oriented either perpendicularly or parallelly to the pump polarization. The dependence is plotted in terms of the pump polarization angle φ_p . This configuration is the same as chosen for the experiments in Fig. 3c of the main text. B) The same theoretical dependence of polarized SPDC rate as in a), however now evaluated in terms of the analyzer polarization angle φ_{pol} . This configuration corresponds to the setting chosen for many polarized SHG measurements from TMDs.

6. Polarization resolved measurements: in Fig. 3a some SHG data point look pretty off compared to the fit, in principle this measurement should be easy to do also with more data points.

The reviewer has a very good point here, some of the data points for the SHG measurement are indeed not following the prediction as closely as they should. We also admit that the number of sampling-points could be higher. In principle, this is an important measurement, as the SHG polarization dependence is a first probe for the nonlinear tensor of the material, which then in turn enables the generation of polarization entanglement. In order to remove any doubts regarding the uniformity of the nonlinear tensor and quality of our sample, we follow the reviewer's suggestion and repeat this measurement with a higher number of data-points and nominally unchanged settings. The new measurement with a higher sampling rate is plotted in Figure 7 and shows now a very close agreement of the SHG measurement with the theoretical expectation. We show this new graph now in Fig. 3a of the main manuscript.

Figure 7: Result of new measurement for polarization-dependent SHG for co-rotating SHG pump and analyzer in parallel configuration. These results are inserted into panel a) of Fig. 3 of the main text.

Major Revisions

1. Fig S1d, the theory doesn't actually match with Ref. [5], could the authors clarify why? From the black "theory" curve shown in Fig. S1d the coherence length at 1550 nm seems to be ~ 800 nm, while it has been shown that is around 550 nm. Please clarify.

We thank the reviewer for this remark and will detail this in the following. The coherence lengths in the theoretical curve in our Fig. S1d and the results of Ref. [5] differ as the considered interacting wavelengths in the nonlinear process are different. For the considered second-harmonic generation (SHG) Ref. [5], the fundamental wavelength is $\lambda_{FF} = 1520$ nm (0.815 eV) and the corresponding second-harmonic wavelength $\lambda_{SHG} = 760$ nm (1.630 eV). Based on the refractive index measurements reported in [5], this corresponds to $n_{FF} = 3.795$ and $n_{SHG} = 4.512$, respectively. From there, a wave-vector mismatch of $\Delta k = k_{SHG} - 2k_{FF} = \frac{4\pi}{\lambda_{FF}}(n_{SHG} - n_{FF}) = 5.924 \cdot 10^6 \text{m}^{-1}$ and correspondingly a coherence length of $L_c = \frac{\pi}{\Delta k} \approx 530$ nm can be computed, which is in agreement with the experimental findings in Ref. [5]. In our SHG measurements, we used a fundamental wavelength of $\lambda_{FF} = 1576$ nm and correspondingly a second-harmonic wavelength of $\lambda_{FF} = 788$ nm.

These wavelengths were chosen to probe the reversed case of SPDC pumped at $\lambda_p = 788$ nm. Based on the refractive index measurements in Ref. [5], the indices for these wavelengths are $n_{FF} = 3.92$ and $n_{SHG} = 4.39$, which leads to a wave-vector mismatch of $\Delta k = 3.732 \cdot 10^6 \text{m}^{-1}$ and correspondingly to a coherence length of $L_c \approx 840$ nm in our case. This corresponds closely to the value mentioned by the reviewer based on our experimental measurements for the thickness-dependent SHG efficiency. It is also worth noting that the theoretical curve of the SHG efficiency is a multiplication of the phase-matching curve, which peaks at L_c , and a modulation by Fabry-Perot interference stemming from the high index contrast between the TMD and the surrounding medium. The exact derivation is detailed in Ref. [5]. Due to this modulation of the phase-matching curve, depending on the exact used wavelength combination the maximum SHG efficiency might not be observed at L_c , unless it directly coincides with a maximum of an FP interference fringe. In sum, we therefore think that there is no disagreement between our data and the results reported in Ref. [5].

To better highlight and explain our choice of wavelengths, we insert an additional sentence at the respective point in the supporting information, stating:

This wavelength is chosen to correspond to the reverse process of degenerate SPDC excited with a pump laser at $\lambda_p = 788$ nm.

Furthermore, we explicitly add the value of the coherence length in the supporting information and the main text.

SI:

... From there we conclude, that our sample with a thickness of $t=285$ nm reaches about 3% of the theoretical maximum SHG efficiency, which is reached at $t \approx 800$ nm. This is close to the coherence length of $L_c \propto 840$ nm for SHG pumped at 1576 nm. ...

Main text:

We however note, that this crystal thickness does not correspond to the global maximum of conversion efficiency. This would be reached for a thickness of $t \approx 800$ nm, close to the coherence length of $L_c \approx 840$ nm for the conversion process excited at 1576 nm.

2. Along the previous comment, how did the authors choose the thickness of the flake in the present work? And why not choosing a flake that has a thickness close to the “constructive” peaks in the Fabry-Perot resonances?

We thank the reviewer for this important comment. In short: we are aware that the thickness of our 3R-MoS₂ sample is (unfortunately) indeed far from ideal in terms of maximizing the SPDC generation rate. This is readily seen in the SHG thickness analysis of Fig. S1d in the supplementary information. The two main reasons for this situation are the limited thickness-control in our fabrication-method based on mechanical exfoliation and the restrictions in the wavelength range based on the detectors and excitation lasers we have available. However, none of these points is a fundamental issue for the use of 3R-MoS₂ as entangled photon sources, which we will describe in more detail in the following.

During our many exfoliation runs of 3R-MoS₂ we have noted, that exfoliated crystals with sufficiently large cross-section either ranged mainly below 300 nm thickness or were much thicker in the range of several microns. In Figure 8 we show microscope images overlaid with height measurements of five exemplary 3R-MoS₂ flakes we exfoliated. These show the described tendency of obtaining crystals with <300 nm thickness in exfoliation. The crystal we used for all measurements in the paper is shown in panel a). We selected this sample, since it has a much larger plateau with uniform thickness than all other crystals we obtained. As detailed in the main manuscript, this is important for the generation of maximally entangled quantum states, since any crystal borders, strain, cracks etc. might distort the nonlinear tensor. From the calculations we present in the manuscript it is however clear, that only a fully symmetric tensor can generate the demonstrated range of entangled states. Therefore, we tried to avoid measurements close to any obvious crystal imperfections. This is also the reason why we didn't try to perform measurements on very small “islands” with higher thickness that are closer to the optimum height value.

Figure 8: a)-c) Height measurement of several exfoliated 3R-MoS₂ crystal. The sample shown in a) is the one investigated in the manuscript. The seemingly “overexposed” region in the bottom right corner of b) corresponds to a region with thickness exceeding 3.5 μm .

Given this observed behaviour of exfoliation, we had to do a trade-off between a sub-optimal thickness regarding the generation rate but otherwise pristine crystal quality that allows the generation of maximally entangled states. Since the aim of our work was not to claim an efficiency record but rather to demonstrate the entanglement properties of the generated states, we believe that the overall findings of the work are not compromised by this. This is in particular true, since it has already been shown in several works that established nanostructuring techniques can also be used to nano-pattern bulk 2H-MoS₂ (e.g. M. Nauman et. al., Nat Commun **12**(1), 5597 (2021), Ref. 39 in manuscript) and also 3R-MoS₂ (e.g. G. Zograf, et. al. arXiv:2308.11504 (2023)). We will include the latter preprint as additional reference into the manuscript. With this, scalable device fabrication of any thickness is possible, which makes it the preferred route over manual exfoliation for future works.

On a side note: it is possible, that we are not the first ones to notice this behaviour of exfoliation for 3R-MoS₂. The authors of Nat. Photon. **16**(10), 698–706 (2022) (Ref. 13 in main text, ref. 5 in SI) may have experienced a similar effect when exfoliating 3R-MoS₂, since in their study of the thickness-dependent SHG efficiency in 3R-MoS₂ they do have a lot of -points for samples with thickness <300 nm and only a very sparse set of datapoints for crystals with larger thickness (compare Fig. 2b). This is of course a speculation which we cannot fully confirm.

We did not include this full discussion on the exfoliation behavior of 3R-MoS₂ into the manuscript, since we did not carry out a full parametric study that would be needed for reproducibility by others (different types of adhesives, exfoliation speed etc.). Based on the reviewers remark we however did realize that we need to comment more on this point already in the main text. Therefore, we did include the following statement into the main manuscript:

...The signal yield from this crystal exceeds a ML-MoS₂ by more than three orders of magnitude. We however note, that this crystal thickness does not correspond to the global maximum of conversion efficiency. This would be reached for a thickness of ≈ 800 nm, close to the coherence length of $L_c \approx 840$ nm for the SHG conversion process excited at 1576 nm. At this thickness, the SHG and by correspondence, also SPDC efficiency would be further increased by a factor of 33x (see supplementary Fig. S4 and supplementary note S3). While we did not reach this optimum thickness with the limited control offered by mechanical exfoliation, nanofabrication techniques that allow precise thickness control and nanopatterning of (3R-) MoS₂ have already been demonstrated [33,34] and can be used for future sample fabrication. ...

3. In Fig. 4b the authors show the idler/signal spectrum, transmitted/reflected through/by a dichroic mirror with cutoff around 1550 nm. Why are these spectra so different than the one shown in Fig. 2d? My understanding is that in Fig. 2d signal and idler spectra are not separated, however the spectrum does not extend beyond 1680nm, while in Fig. 4b it seems that the idler spectrum extends to 1900nm. With a 788nm pump the longest wavelength you can detect coincidences with is ~ 1660 nm due to energy conservation. Can the authors motivate this choice?

This is an important remark that made us realize that we indeed need to explain the method of fiber spectroscopy using correlated photons in more detail already in the manuscript. A similar point has been raised by reviewer #2, question 3.

The difference between the spectra arises because for the measurements reported in Fig. 2 and 3 a long-pass filter with cut-on wavelength 1500 nm is inserted in the SPDC beam-path, while for the measurements in Fig. 4 this filter was removed. However, unfortunately we missed to explicitly state the removal of this filter, which may have caused the confusion.

The measurements related to Fig.2 and 3 all use a non-polarizing fiber beam-splitter with central operation wavelength 1550 nm and a bandwidth of approx. ± 100 nm (Thorlabs TW1550R5F1). The main purpose of the longpass filter with cut-on 1500 nm is to limit the SPDC spectrum to a range well within the operation range of the fiber beamsplitter. We had already mentioned this in the method section, however, now also briefly bring this point up in the main text.

In the experiment, the spectrum is limited only by the long-pass filter with cut-on 1500 nm which is used to limit the SPDC spectrum to the operating bandwidth of the fiber beamsplitter and for suppression of residual photoluminescence ...

We also explicitly marked the 1500 nm longpass filter now in the schematic of the experimental setup in Fig. 2a and mention the model number of the fiber beamsplitter in the method section.

For the quantum state tomography, we define the signal and idler frequency modes by replacing the fiber beamsplitter with a dichroic mirror as described in the main text. This has a much larger operating bandwidth than the fiber-splitter, which is why we could remove the 1500 nm long-pass filter. We add this as an explicit statement in the main text:

The previously used long-pass with cut-on wavelength 1500 nm is removed now, since the operating bandwidth of the dichroic mirror is larger than for the fiber beamsplitter.

Since the SPDC spectrum from the non-phased-matched crystal is expected to be extremely broad and considering that the 1500 nm long-pass filter was not inserted for the measurements in Fig. 4, the limiting factor for the SPDC spectrum is now only the detection range of our experimental setup.

The super-conducting nanowire detectors (SNSPD) we use are optimized for detection at 1550 nm and slowly drop in efficiency for wavelengths much larger than this. Since the spectral measurement is based on correlations between signal and idler, this limit on the long-wavelength side of the spectral detection simultaneously defines the edge of the short-wavelength side of the spectrum. We briefly add this explanation in the main text:

Here, the width of the observed SPDC spectrum is mostly limited by the detection range of our experimental setup where the efficiency of the used SNSPDs slowly drops towards wavelengths much longer than their optimized operation wavelength at 1550 nm. Due to the correlation-based spectral measurement method, this simultaneously limits the short-wavelength side of the spectrum.

Since both reviewer #1 and #2 remarked on the spectral measurement, we also got the impression, that we should slightly extend the description of the correlation-based method with the long dispersive medium in general. Accordingly, we added an explanatory sentence

Furthermore, we measure the SPDC spectrum using fiber spectroscopy [35], where the photon pairs are first sent through a long, dispersive medium before coincidence detection. In our experiment we insert 1km of SMF28 single-mode fiber before the beamsplitter for the spectral measurement. After propagation through this medium with known dispersion, the arrival time difference between both photons can be mapped to their wavelength difference (see methods for more details).

4. One of my major concerns is the value of the CAR. It is extremely low, making this system practically useless. Could the authors compare their CAR to typical CAR values of standard SPDC sources like e.g. BBO, PPLN, PPKTP? What are the minimum requirements in real-world applications? How do you envision the use of TMDs in the future given such low performances? Complementing the manuscript with a short discussion about the state of the art of SPDC generation, and a comparison with this work, would make the paper stronger, and more accessible also to readers who are not necessarily working in the field of 2D materials.

The reviewer raises here several very important points. As a very short first answer: the CAR of an SPDC source is not a material parameter like for instance the nonlinear coefficient or symmetry of the nonlinear tensor. Therefore, we cannot directly compare the CAR of a 3R-MoS₂ SPDC source to one based on ppLN, ppKTP etc. While the CAR of our “source” in this proof-of-principle experiment is indeed low, this is primarily rooted in the sample being a crystal with sub-wavelength thickness on a bare substrate (i.e. without embedding into a cavity, nanostructuring etc.). The material itself is very well suited for a source with high CAR (or in different terms: heralding efficiency). Here, we didn’t design a readily optimized system. Despite that lack of system optimization, we show high-fidelity entangled states and the tuning thereof, which is different from most other materials where also the generation of entanglement would require carefully optimized systems (Sagnac-interferometers, displacement interferometers, sandwiched crossed-crystals, heating systems for phase-matching via temperature tuning etc.). We will discuss this now in detail. However, this is a quite broad question, which is why we first need to clarify a few key points about application-ready SPDC sources before putting the material 3R-MoS₂ into the picture.

Let us first briefly recap the important properties of a technologically useful SPDC source, for example for a quantum communication network. Here, the CAR is only one out of three important points. Following the reviews of A. Anwar, et. al., Review of Scientific Instruments 92, 041101 (2021). (Ref. 5 in main text) and A. Anwar, et. al., “Development of compact entangled photon-pair sources for satellites”, Applied Physics Letters 121, 220503 (2022) (Ref. 10 in main text), we need to look at three key aspects: entangled photon-pair rate, CAR (or heralding efficiency) and state fidelity.

- Entangled photon-pair rate.** Here it is important to distinguish between several quantities. The first benchmark is, the “generated pair-rate” or “generated brightness”, i.e. the total number of pairs per time and unit pump power created in the nonlinear medium (unit: Hz/mW). The second quantity is the “detected pair-rate” or “detected brightness”. This is defined as the number of pairs that are registered by the detectors as coincidence counts (unit: Hz/mW). This is the technologically more relevant parameter and is in general much smaller than the total generated pair-rate due to collection losses, limited detector efficiency etc. Lastly, applications usually require pairs generated in a certain limited bandwidth, for instance within the telecom band. Therefore, also the “detected *spectral* brightness” is of importance (unit: Hz/mW/nm). Here we already see that the nonlinear medium is only one ingredient for achieving a high detected (spectral) brightness. Equally important are collection efficiency, the method of spectral shaping (phase-matching, cavities, filters) and the efficiency and timing resolution of the detectors. State of the art bulk sources based on PPLN, PPKTP, BBO etc. with macroscopic foot-prints range somewhere between 10^0 - 10^5 Hz/mW/nm, greatly depending on source design, used detectors, spectral band etc., see (Ref. 5 in main text). The detected rate in our experimental setup is $\approx 1.2 \cdot 10^{-2}$ Hz/mW, however the generated rate is difficult to estimate due to the poor collection efficiency from an ultra-thin crystal (see estimation of the collection efficiency in the methods section, which has to be regarded as very conservative.) Based on our estimate with the NA=0.4 lens, the generated rate is about 1.5 Hz/mW in the <300 nm crystal. Due to the very broadband spectrum, the spectral brightness is very low in these sources.
- Coincidence-to-accidental ratio, or Heralding efficiency:** In the paper, we measure the coincidence-to-accidental ratio CAR as a sign of non-classicality of the photon emission. This is defined as the ratio of true coincidences R_C (corrected for accidentals) to accidental coincidences R_{acc} such that $CAR = \frac{R_C - R_{acc}}{R_{acc}} = g^{(2)}(\tau = 0) - 1$, and is therefore directly related to the normalized second-order correlation function at zero time-delay $g^{(2)}(\tau = 0)$. An accidental coincidence arises from combinations of all possible correlations of actual signal/idler photons (s/i), detector darkcounts (dc) and background photons (bg) (which might e.g. be photoluminescence, straylight in the lab etc.). These combinations could for instance be correlations of two background photons bg/bg; two darkcounts dk/dk; signal/idler with background etc. Please refer to L. Marini, et. al. JOSA B, 35, 4, p. 672, (2018) (Ref. 21 in main text) for a more complete analysis of all possible cases to generate an accidental coincidence and losses in a correlation detection setup for SPDC. What becomes immediately clear from here: the CAR of a certain SPDC source depends on many parameters of the entire system like the efficiency of pair collection, detector efficiency and dark-counts, background light etc. and therefore on the entire system design of the source rather than only on the nonlinear material. Application-ready SPDC sources are often characterized in terms of a slightly different quantity, namely their signal and idler heralding efficiency $\eta_{1,2}$. This is defined as $\eta_{1,2} = \frac{R_C - R_{acc}}{R_{1,2}}$ where $R_{1,2}$ are the single count-rates in the signal and idler channel. This quantity is related to CAR via $\eta_{1,2} = CAR \times R_{1,2} t_c$ where t_c is the coincidence time-window. Typical bulk sources have heralding efficiencies between a few percent to over 50%. For our source, we estimate a heralding efficiency on the order of $2 \cdot 10^{-5}$, which is limited in an ultrathin crystal by the small collection efficiency and residual photoluminescence.
- State fidelity:** This is a measure how close the state emitted by the source is to a target state, typically a pure, maximally entangled Bell state. We explained and derived this part in detail in the SI. Typical sources based on PPLN, PPKTP, BBO etc. reach fidelities > 90%, often better than 95%. In our source we find a state fidelity >95%.

First of all and importantly: the Bell state fidelity we demonstrate is competitive with highly optimized “state-of-the-art” sources. We want to highlight this point again, since changing the degree of quantum entanglement after generation would require lossy transformations (S. Lung et al., ACS Photonics, 7, 11, pp. 3015–3022, 2020) which is generally undesirable for an efficient source and would be a fundamental problem for 3R-MoS₂.

Comparing further the CAR and detected rate with optimized sources, we see that the values we measured are much lower. This however is not a fundamental issue with the material 3R-MoS₂ since both detected rate and CAR crucially depend on the design and optimization of the entire system. What we have presented here is a proof-of-principle experimental setup, which is by no means an optimized SPDC source. To the contrary: if the objective is to optimize spectral brightness and CAR, using an extremely thin nonlinear crystal on a bare substrate (i.e. without embedding into a cavity, nanostructuring etc.) is close to the least optimal configuration for several reasons:

- **Pair-emission into very large angular range:** due to the lack of longitudinal phase-matching, the emission cone of photon-pairs is very large and is expected to exceed the collection numerical aperture by far. In addition, pairs can be emitted in different directionalities entirely, namely forward-forward, forward-backward and backward-backward (where forward means in the same direction as pump propagation and backward against the direction of pump propagation). In our experiment, we can only collect forward-forward pairs in a narrow solid angle and miss all other pairs. The detected pair-rate in our experiment is therefore much smaller than the generated rate. This is discussed in detail e.g. in L. Marini, et. al. JOSA B, 35, 4, p. 672, (2018) (Ref. 21 in main text). Furthermore, we detect photon-pairs through single-mode fiber coupled detectors, while the emission from a sub-wavelength-thickness crystal contains many spatial modes. The collection efficiency through a single-mode fiber is therefore low in such a system (see for instance the supplementary material of T. Santiago-Cruz et al., Science, 377, 6609, 991–995, (2022) Ref. 42 in main text for a discussion).
- **Very broad spectrum:** due the lack of phase-matching, also the frequency spectrum is very broad (see our measurements). Given inevitable chromatic aberrations of all used lenses and the limited bandwidth of the used single-mode fibers and detectors, this intrinsically limits the number of detected pairs. Furthermore, in the presence of broadband noise processes, e.g. photoluminescence, frequency filtering for noise suppression is difficult if also the SPDC bandwidth is extremely broad.
- **Competition between coherent nonlinear generation and incoherent noise processes:** as we pointed out in the main manuscript, photoluminescence from TMDs can be a large noise contribution in correlation-based SPDC measurements to the point that it prevents detection of SPDC at all (see e.g. H. Dinparasti Saleh, et. al., Scientific Reports 8, 7842 (2018), Ref. 22 in main text). Therefore it is important that the indirect bandgap of 3R-MoS₂ drastically reduces PL emission as compared to e.g. monolayer MoS₂. We confirmed this in our manuscript by measurements, where we could not discern a PL signal above the detector noise for wavelengths $\lambda > 1300$ nm, as measured with an InGaAs chip in our spectrometer. However, we suspect that there is still some PL emission visible to the considerably more sensitive superconducting nanowire single photon detectors we use for the SPDC measurements, which therefore reduce the CAR. When evaluating this “competition” between SPDC and PL signal, it is now important to consider their scaling with the sample size. The photoluminescence rate R_{PL} , being an incoherent process, scales linearly with the sample length L ($R_{PL} \propto L$) whereas the SPDC rate R_{SPDC} , being a coherent process, scales quadratically in L ($R_{SPDC} \propto L^2$ for a phase-matched process).

Therefore, noise from PL from the nonlinear material mostly affects thin samples but quickly reduces in relative strength compared to SPDC for thicker samples due to their different scaling behaviors (see e.g. V. Sultanov and M. Chekhova, ACS Photonics, 11, 1, pp. 2–6, (2024) for more details. We added this reference now to the main text).

Having all these points in mind, the strategy to design an application-ready SPDC source from 3R-MoS₂ is quite clear. A first and straightforward option is to increase the sample thickness to enter a regime of (quasi-)phasematched SPDC generation. This will simultaneously improve the source brightness and the CAR ratio. The longer interaction length leads to higher pair-rates ($R_{SPDC} \propto L^2$), more spatially directed pair-emission and in general suppression of counter- or backward propagating pairs (equals higher collection and heralding efficiency), narrower SPDC spectrum and relative reduction of PL compared to SPDC ($R_{PL} \propto L$ vs. $R_{SPDC} \propto L^2$). We had already mentioned quasi-phasematching (QPM) via stacking of rotated TMD crystal sheets in our discussion as a promising pathway to increase generation efficiency. During the review phase of our work, a preprint showing an experimental realization of quasi-phasematched nonlinear conversion and SPDC in 3R-MoS₂ has been published (C. Trovatiello et al., ‘Quasi-phase-matched up- and down-conversion in periodically poled layered semiconductors’. arXiv, Dec. 31, 2023. doi: 10.48550/arXiv.2312.05444, we cite this work now). As expected, already a small number of quasi-phasematching periods leads to a strong increase of pair-rate and CAR (in their work CAR=37). Importantly, the symmetry of the nonlinear tensor is preserved, such that this QPM scheme is fully compatible with our presented way of generating entanglement. With this, all previously developed strategies to optimize pair-sources based on ppKTP, pplN etc. can be immediately used for 3R-MoS₂. The advantage is that 3R-MoS₂ offers the same conversion efficiency over much shorter device lengths (higher nonlinearity) and intrinsically creates entanglement (see our work). Therefore, the introduction of 3R-MoS₂ into existing SPDC source concepts is rather straightforward and yields immediate benefits. In the conclusion section of the main paper, we also mention several other pathways to increase generation efficiency which all will automatically improve the CAR as well, e.g. integration with cavities and nanostructuring. The latter are currently ongoing projects, however discussing design details in this direction would go beyond the scope of the current paper.

We add a short summary of this discussion to the conclusion section of the manuscript:

Schemes like quasi-phasematching and cavity integration which result in narrower spectral and spatial emission would not only enhance the generation rate but also improve the photon-pair collection efficiency. Additionally, an enhanced SPDC efficiency for instance through longer interaction lengths, proportionally reduces the effect of photoluminescence [45]. This, together with the enhanced generation and collection efficiency, will improve the coincidence-to-accidental ratio in our source.

...

Combined with the avenues for scaling the generation rate, coincidence-to-accidental ratio and pair collection efficiency, this gives TMDs a clear advantage as nonlinear material platform for entangled photon-pair sources.

5. The CAR values in S4.1 do not really follow the expected dependence $CAR=1/P_{\text{Pump}} + 1$. Why? The behavior looks linear to me.

The reviewer is correct, from our measurement of the CAR at different pump powers, a specific functional dependence cannot unequivocally be deduced. We measured the CAR for these pump powers to prove that we indeed measure photons generated by the SPDC process, which results in a CAR larger than 1. This is equivalent to a second-order correlation function at zero time-delay $g^{(2)}(0) > 2$ since both quantities are simply related by $CAR = g^{(2)}(0) - 1$. Please also refer to reviewer #3, answer 4 for a short discussion of $g^{(2)}(0)$ and CAR. $g^{(2)}(0)$ values larger than 2 are not possible with (classical) thermal or coherent light, and hence are a hallmark of the SPDC process.

Due to the low count rates in our experiments, the errors in the measured CAR values are significant and an accurate assessment of their dependence on the pump power is accompanied by many inaccuracies. A deviation from the functional dependence mentioned by the reviewer can occur especially at low count rates, where the detector dark-count rates become comparable to the detected photon rates due to SPDC. For such low rates, the CAR is initially increasing with increasing pump power until reaching a maximum, then it decreases again following the dependence given by the reviewer [K.I. Harada et al., Frequency and polarization characteristics of correlated photon-pair generation using a silicon wire waveguide, *IEEE Journal of Selected Topics in Quantum Electronics* 16, 325 (2009), A.S. Clark et al., "Heralded single-photon source in a III-V photonic crystal," *Opt. Lett.* 38, 649-651 (2013)]. In our experiments, we are in this regime of low SPDC count rates, and thus a different dependence is not unexpected.

In short, I believe this work is novel and impactful, and I believe it is of broad interest in the field. I'd be happy to support publication if the authors implement the revisions above.

Reviewer #2 (Remarks to the Author):

The manuscript entitled “A Tunable Transition Metal Dichalcogenide Entangled Photon-Pair Source,” by Weissflog and co-workers report on the experimental demonstration of entangled photon pair generation via spontaneous parametric down conversion (SPDC) in a 285-nm-thick flake of MoS₂ in the 3R phase (which exhibits no inversion symmetry and, thus, second order nonlinearity). The authors first theoretically show that the symmetry of the crystal (point group C_{3v}) straightforwardly allows for the generation of maximally entangled photons in two Bell states, as well as for polarization tunability without compromising the entanglement. Then, they experimentally confirm entanglement and polarization tunability, as well as observe the two predicted Bell states. In my opinion, the work is novel, timely and can be of interest to the wide scientific community working in the fields of quantum technologies, nonlinear optics and layered (2D) materials. The work seems to have been carried out with proper care, and the manuscript is clear and accurate. I would recommend its publication in Nature Communications after the authors address the following major issues:

1. Even though it is indeed interesting that Bell states can be directly obtained by simple pump polarization control, this is a property of any crystal in the D_{3h} (or, within reasonable approximation, the C_{3v}) point group. Hasn't this been previously explored even for bulk crystals with the same symmetry? If it has, please add references.

The point raised by the reviewer here is of course correct and indeed very interesting. As mentioned several times in the manuscript, the generation of entangled states in TMDs is closely connected to the crystal symmetry of the D_{3h}, or here the C_{3v} point group. As correctly pointed out by the reviewer this means, that in principle any material with this crystal symmetry should be able to generate entangled photon-pairs and more broadly speaking, should have the same polarization properties as the 3R-TMD investigated in this work for nonlinear conversion. Following the same intuition as the reviewer, after realizing that the C_{3v} tensor offers these interesting properties, the first thing we checked was if something along similar lines had been reported in the literature before. However, also after an extensive search, we were not able to find any proposals, much less demonstrations along the line of photon-pair generation that uses the D_{3h} or C_{3v} tensor in this way, neither in bulk nor thin crystal systems. This is still somewhat surprising to us, particularly when considering that technologically very relevant nonlinear materials like lithium niobate (LN) or beta barium borate (BBO) have C_{3v} symmetry. These materials are very frequently used for pair-generation via SPDC, however in a different configuration than in our approach here. Conventionally, sources for nonlinear wave generation in BBO and LN are operated such that (quasi-) phasematching between the pump signal and idler waves is achieved. For this, one either needs birefringence or can resort for example to periodic poling. Both approaches are heavily used in nonlinear sources, for a very comprehensive overview of different source designs with these materials see for instance A. Anwar “Entangled photon-pair sources based on three-wave mixing in bulk crystals,” *Review of Scientific Instruments* 92(4), 041101 (2021) (Ref. 10 in main text). If the reviewer is aware of any references where the crystal symmetries have previously been exploited in a similar way for photon-pair generation in other materials, we are very happy to add them to the manuscript.

The reviewer comment motivates us to mention more explicitly, that in principle also other materials with this symmetry group can be used for entangled state generation in the demonstrated way.

We therefore add the following statement to the “conclusion” section:

This decoupling of entangled state tuning from the generation efficiency results in a highly flexible and easy to operate, tunable entangled photon-pair source. Since all these properties are directly derived from the crystal symmetry, no external optical components like interferometers etc. are needed for generating entanglement. This is the simplest conceivable, tunable entangled photon-pair source, a prerequisite for active quantum networks, which enable for instance multi-user quantum secret sharing [11]. Furthermore, this direct link to the crystal symmetry allows to generalize our results for tuneable entanglement generation to other nonlinear materials of the same or similar symmetry group, for instance lithium niobate or beta barium borate (BBO).

As a last comment we would like to add that even though other materials like LN or BBO have the same crystal symmetry, 3R-MoS₂ is still a highly desirable platform to realize these effects in. The reasons are the higher nonlinear coefficient than LN and especially BBO, high refractive index that allows strong field confinement, easy integrability with other material platforms and the possibility to create quasi-phase-matching along the symmetry axis. The latter is mainly enabled by the atomically smooth interfaces of the van-der-Waals material which allows stacking of the material without introducing losses, which is hard to achieve in other materials. For more details on this point see C. Trovatiello, et. al., Quasi-phase-matched up- and down-conversion in periodically poled layered semiconductors, arXiv:2312.05444 (2023), Ref. 27 in main text for details).

2. Along the same subject, the authors show that 2 out of the 4 states of the Bell basis can be directly obtained. They should comment on how the other 2 states can be obtained without significantly increasing the complexity of the system.

In principle, all 4 Bell states can be generated by linear single-qubit operations from one of the Bell states. For states in the polarization basis, as generated in our experiments, realizing these transformations involves deterministic spatial splitting of the signal and idler modes, the application of wave plates on the separated modes and their recombination using a symmetric beam splitter [S. Mishra, R.P. Singh, Transformation of Bell states using linear optics. Physics Open 18, 100199 (2024)]. To be applicable to our source, the signal and idler photons should not be separated by their frequency, as is done in our experiment, since this would impede recombination and mixing of these modes using a simple beam splitter. Alternatively, signal and idler could be separated using the spatial degree of freedom exploiting momentum conservation.

Compared to the simplicity of the demonstrated source for two different Bell states, any manipulation stage significantly increases the complexity of the complete optical arrangement. Importantly, the two generated states are sufficient to cover several important applications of Bell states. In quantum communication, often states in the form $|HV\rangle + |VH\rangle$ are used. On the other hand, the state $|HH\rangle - |VV\rangle$ can be easily transformed into a NOON-type state for applications in quantum sensing using a polarization beam splitter.

3. The spectrum in Figure 4b is significantly broader than that in Fig. 2d. Why is that? I assume that in the short wavelength edge the reason might be that the 1500nm cut-on filter is not being used (although this must be explicitly stated somewhere). But why does the long wavelength edge also stretch further in this figure?

We thank the reviewer for pointing out this aspect. As the reviewer correctly assumed, we had removed the long-pass filter with 1500 nm cut-on wavelength for all measurements reported in Fig. 4. Unfortunately, we missed to mention this part clearly in the manuscript, a mistake that we corrected now. Also reviewer #1 had noted this problem in “major revision 6”, where we already gave a detailed explanation. Since the questions of both reviewers are very similar, we only give a short version and would also refer to the previous answer for more details.

In summary, for all SPDC measurements we reported in Fig. 2 and 3, we used a single fiber-coupling port for both photons and a subsequent fiber beamsplitter which has a centre operation wavelength of 1550 and a bandwidth of approx. ± 100 nm (Thorlabs TW1550R5F1). We used the long-pass filter with cut-on 1500 nm here to restrict the band-width of the photon pairs well within the working range of the fiber-splitter. Since this was not necessary for the more broad-band free-space dichroic beamsplitter used for the measurements in Fig. 4, we removed the long-pass filter in these measurements. We noted this now explicitly in the main manuscript and also marked the filter explicitly in Fig. 2a, see answer to rev. #1, qu. 6.

The reason that also the long-wavelength edge extends further in the measurement in Fig. 4b is that we use a correlation-based fiber-spectrometer here. In this, the arrival time difference between two photons of a pair after they propagated through a long dispersive medium is mapped to their wavelength difference. This also means, the wavelength of a pair can only be measured if actually both photons are detected. Therefore, placing a filter which blocks one of the two photons defines the detectable two-photon spectrum via energy conservation. One spectral edge will be at the filter cut-off wavelength while the second edge is at its conjugate wavelength. Therefore, removing the short-pass filter at 1500 nm simultaneously extends the long-wavelength side of the spectrum, where pairs can be detected via correlations. The new limiting factor is the operation range of the detectors, see the previous answer to rev. #1, qu.6 for more details.

To make this point clearer, we add the following explanation in the method section on fiber spectroscopy:

Note that in this correlation-based measurement, always both photons of a pair need to be detected. An edge-pass filter therefore determines via energy conservation the entire width of the detected photon-pair spectrum. For instance in the spectrum measured in Fig. 2(d) using long-pass filter with cut-on wavelength $\lambda_c = 1500$ nm and pump wavelength 788 nm, the long-wavelength edge of the spectrum is therefore at $\left(\frac{1}{\lambda_p} - \frac{1}{\lambda_c}\right)^{-1} = 1660$ nm.

4. In the conclusions' section, the authors state: "we show that for any linear pump polarization, a different, maximally entangled state is generated while the generation efficiency is independent of the pump polarization." They must clearly say that this is a theoretical result, as they only show experimental results with the pump polarized along x and y.

This is correct and we agree that we have to be more precise at this point. We therefore reformulated the statement in the following way:

We demonstrate that TMDs intrinsically generate maximally entangled polarization Bell states. Experimentally we show this for two different pump polarizations and then further theoretically derive that in fact for any linear pump polarization, a different maximally entangled state is generated while the generation efficiency is independent of the pump polarization.

5. Also in the conclusions, the authors say that a possible way to increase the source brightness would be to exploit exciton resonances. But couldn't this reduce quantum coherence? Please comment.

For classical nonlinear frequency conversion in TMDCs it is well documented, that the nonlinear susceptibility is strongly dispersive and resonantly enhanced by excitonic resonances in the TMDCs. In particular for second-harmonic generation, the enhancement is associated to excitons at the second-harmonic wavelength. Since SPDC is governed by the same physical rules as second-harmonic or sum-frequency generation, a similar enhancement of the nonlinear coefficient can be expected, leading to an enhanced pair-generation rate. Since the excitons are excited by the shorter pump wavelengths, this would not lead to additional losses for the signal or idler photons, thus keeping the two-photon component of the generated state intact. However, an effect that remains to be studied in detail is the emission of additional single photons due to increased fluorescence of the TMDCs, which may be present for pump wavelengths at or even below the excitonic resonances.

We would also like to note that systems exhibiting excitons, namely semiconductor quantum dots, have been already shown to enable the generation of entangled photon pairs [O. Benson et al., Phys. Rev. Lett. **84**, 2513(2000), Y. Chen et al., Highly-efficient extraction of entangled photons from quantum dots using a broadband optical antenna, Nat. Comm. 9, 2994 (2018)].

6. In Methods – Sample Fabrication: a reference must be given to the crystal synthesis method, and the substrate pre-treatment must be described.

We thank the reviewer for pointing this out and modify the related paragraph in the method section accordingly:

Bulk 3R-MoS₂ crystals were grown using the chemical-vapor transport technique.¹ Subsequently, 3R-MoS₂ flakes were prepared on polydimethylsiloxane (PDMS), which begins with mechanical exfoliation of the crystals. Afterwards, the substrates were pre-treated by oxygen plasma, in order to eliminate potential contamination and improve the adhesion, followed by a dry transfer method to transfer the 3R-MoS₂ flakes onto quartz substrates.

We add the following reference for the crystal synthesis method:

Zhao, M. et al. Atomically phase-matched second-harmonic generation in a 2D crystal. *Light: Science & Applications* **5**, e16131-e16131 (2016).

7. In Methods - Polarization-Resolved SHG Measurements and SHG Mapping: the authors state that “Polarization-resolved SHG measurements were carried out with the same setup as used for quantum measurements but working in reverse: the fundamental beam was incident from one of the collecting fibers and[...]” Does this mean that the femtosecond pulse passes through 1 km of single mode fiber? This would significantly alter its temporal and spectral characteristics. I believe this is not the case. Please clarify.

The reviewer is of course right that we are not sending the SHG pump pulse through a 1 km fibre. Unfortunately, we didn't state this clearly in the manuscript yet. Therefore, we update the related paragraph of the methods section:

As a laser source, a tunable femtosecond laser (Coherent Chameleon with optical parametric oscillator Angewandte Physik und Elektronik GmbH APE OPO-X) with pulse width 100 fs, repetition rate 80 MHz, at a central wavelength 1576 nm and with FWHM 10 nm was used. Note that the pulses were not sent through the normal detector fiber but through a shorter single-mode fiber (Thorlabs SMF-28-J9-CUSTOM) with length 0.5 m to avoid distortion of the pulses.

We also add a comment in the schematic of the experimental setup in Fig. 2a, stating that the 1km SMF fiber is 'optional', as it is not used in all parts of the measurement but only for the spectrum characterization.

8. In Supplementary Note S2, the authors say that the $\chi^{(2)}$ dispersion in the spectral range of interest is negligible; but later they attribute differences in the $\chi^{(2)}$ values measured in references [9] and [10] to a high $\chi^{(2)}$ dispersion. How are these two statements compatible? Even if the $\chi^{(2)}$ dispersion is only significant in the ranges measured in the references, how do the authors know that the d_{16}/d_{31} ratio measured in ref. [9] is maintained at other frequencies ranges, despite dispersion? This is an important issue and must be carefully addressed.

We thank the reviewer for pointing out that this part of the values of the nonlinearities is not clarified enough yet and will address this in detail. We agree that this is a very important point, particularly since the highly dispersive/excitonic behaviour of TMDs, often combined with monolayer thickness of the samples, makes reporting their nonlinearity values a nontrivial task. We also comment on the dispersion of $\chi^{(2)}$ in detail in reply to reviewer #1, minor revision 1. There, in Table 1 we summarized the different reported values for the nonlinearity values of 3R-MoS₂ and the corresponding reference.

Regarding the first part of the question, we want to confirm again that we mean that the dispersion of $\chi^{(2)}$ in our range of interest, i.e. wavelengths larger than the SPDC pump wavelength $\lambda = 788$ nm, is negligible. This is readily seen in the SHG measurement for a wide range of pump wavelengths conducted by Xu. et al. Nature Photonics 16, 698 (2022) (Ref. 13 in main text, Ref. 6 in SI) which we cite at the related point in supplementary note S2. What we referred to with this statement was the difference between the values reported in Refs. 9 and 10 which measure with different excitation wavelengths of 1064 and 1200 nm, respectively. SHG pumped at these wavelengths will then experience a dispersive $\chi^{(2)}$ and therefore yield different values for the nonlinear coefficient.

To avoid any confusion on this point, we reformulate this sentence in supplementary note 2 as:

For conversion from a fundamental wavelength of 1064 nm to a second-harmonic wavelength of 532 nm in 3R-MoS₂, this ratio b is measured in [11] as $b = \frac{|d_{16}|}{|d_{31}|} = \frac{500 \frac{\text{pm}}{\text{V}} \pm 200 \frac{\text{pm}}{\text{V}}}{200 \frac{\text{pm}}{\text{V}} \pm 100 \frac{\text{pm}}{\text{V}}} = 2.5 \pm 1.6$. It is now important to note that at these wavelengths $\chi^{(2)}$ of 3R-MoS₂ is already dispersive [6], which likely is the reason for the difference to the nonlinearity value of $d=0.5\chi^{(2)} = 15 \text{ pm/V}$ measured in [12] (fundamental wavelength 1200 nm, second-harmonic 600 nm). Since we use different wavelengths outside of the dispersive region, these values will not apply exactly to our experiment.

The reviewer's second question regards the ratio between the in-plane and out-of-plane tensor elements which were in Ref. [9] measured for SHG excited at 1064 nm, resulting in a SHG wavelength of 532 nm. Since this SHG wavelength is already in the range where $\chi^{(2)}$ this likely leads to a different absolute magnitude of the nonlinear tensor compared to our wavelength range of interest. While the absolute value of $\chi^{(2)}$ does not affect our calculation of the degree of entanglement, the reviewer is correct that dispersion could potentially also change the relative strength of the nonlinear tensor elements. We currently cannot measure the out-of-plane tensor components and are not aware of any other reference that reports all tensor components of 3R-MoS₂ in our wavelength range of interest. Therefore, we take the values provided in Ref. 9 as a starting point for our calculation and then check, how big the impact of a changed ratio of in-plane and out-of-plane tensor components would be. For this we considered three scenarios of the ratio $\frac{d_{16}}{d_{31}} = b$: $b=2.5$ (default scenario as reported by Ref. 9 Wagoner et. al.), $b=10$ (in-plane tensor components increased) and $b=0.25$ (out-of-plane tensor components much stronger than in-plane tensor components). To further investigate the effect, we also consider different collection numerical apertures of $\text{NA}=0.4$ (same as in our experiment) and $\text{NA}=0.9$. As described in the method section, for these calculations we use a Green's function approach that is fully vectorial and takes into account multiple reflections of pump signal and idler beams (see also E.A. Santos, "Entangled Photon-pair Generation in Nonlinear Thin-films," arXiv:2403.08633 (2024) for a detailed description). In Table 3 we summarize the results for the fidelity $F(\Phi^-)$ with the Bell state $\Phi^- = \frac{1}{\sqrt{2}}(|HH\rangle - |VV\rangle)$ and concurrence C for this set of tensor ratios and NAs considering an x-polarized pump.

Table 3: State fidelity and concurrence for different ratios of in-plane and out-of-plane nonlinear tensor and collection numerical apertures.

		NA=0.4	NA=0.9
$F(\Phi^-)$	b=10	1	0.977
	b=2.5	1	0.976
	b=0.25	0.969	0.894
C	b=10	1	0.954
	b=2.5	1	0.894
	b=0.25	0.964	0.814

As expected, we find as general trend that the state fidelity and concurrence decrease if the strength of the out-of-plane nonlinear tensor elements is increased. However, it is important to note that this effect is only pronounced if the collection numerical aperture is large.

For a moderate numerical aperture of NA=0.4 as used in our experiment, the influence of the out-of-plane components is significantly reduced. To understand this, we have to recall that in the case of our moderately focused Gaussian pump beam propagating along the z-direction, the z-polarized component of the electric field is small. Therefore, tensor components that rely on a z-polarized pump field will practically not contribute to down-conversion. Following our index convention for the nonlinear tensor $\chi_{\alpha,\beta,\gamma}^{(2)}$ where γ refers to the pump polarization, this means that no elements with $\gamma = z$ have to be considered. This leaves the following tensor elements with out-of-plane components: $\chi_{xzx}^{(2)} = \chi_{zxx}^{(2)} = \chi_{yzy}^{(2)} = \chi_{zyy}^{(2)}$. With these elements, either the signal or idler photon will be generated by an out-of-plane nonlinear source, leading to an emission under a large propagation angle. This means, emission via these tensor elements can only be collected efficiently with large numerical apertures. Contrarily, the tensor elements that generate the entangled states $\chi_{yyy}^{(2)} = -\chi_{yxx}^{(2)} = -\chi_{xxy}^{(2)} = -\chi_{xyx}^{(2)}$ only have in-plane components, such that the emission is close to the optical axis. The difference is clearly seen in Figure 9, where we compare the calculated farfield emission patterns for the in-plane and out-of-plane components. While emission from the in-plane elements is concentrated around the z-axis (Figure 9a), the out-of-plane components actually have zero emission along the z-axis (Figure 9b).

Figure 9: Two-photon farfield emission pattern for a) in-plane and b) out-of-plane nonlinear tensor components for an x-polarized pump field. The two detectors for coincidence detection are arranged symmetrically around the z-axis.

This means that irrespective of the ratio between d_{16}/d_{31} , the quantum state emitted along the z-axis will always only be generated by in-plane tensor components. Since the relation of the in-plane elements $\chi_{yyy}^{(2)} = -\chi_{yxx}^{(2)} = -\chi_{xxy}^{(2)} = -\chi_{xyx}^{(2)}$ directly stems from the crystal symmetry, it is not subject to dispersion. With this, the quality of the entangled state generated by 3R-MoS₂ along the z-axis does not fundamentally depend on the ratio of the in- and out-of-plane tensor components which could change with wavelength. This is true even in an extreme case, where the out-of-plane elements are nominally much stronger than the in-plane tensor elements. The degree of entanglement of photon pairs emitted close to the z-axis is the practically important configuration, since the use of longer crystals together with quasi-phasematching, cavities etc. will naturally lead to directed emission close to the z-axis. However, in this work we show experimentally, that also for collection with a moderate numerical aperture of NA=0.4 in practice a highly entangled state can be observed.

In summary we therefore fully agree with the reviewer, that the d_{16}/d_{31} ratio measured in Ref. 9 cannot automatically be assumed to be constant over the whole wavelength range due to dispersion of $\chi^{(2)}$. Currently we cannot determine the dispersive behaviour of the nonlinearity ourselves and are not aware of other references for this. However our analysis showed, that for the moderately focused pumping condition we use, this ratio only affects collection with large numerical apertures. Regardless of the exact ratio of these tensor elements, the state emitted along the z-axis is always fully entangled.

We agree with the reviewer that these points were not formulated precise enough so far in the supplementary document. We therefore modify the discussion in the supplementary information by including the main part of this discussion as well as Figure 9 and Table 3 into supplementary note 2 of the main paper.

As expected, we find as general trend that the state fidelity and concurrence decrease if the strength of the out-of-plane nonlinear tensor elements is increased. However, it is important to note that this effect is only pronounced if the collection numerical aperture is large. For a moderate numerical aperture of $NA=0.4$ as used in our experiment, the influence of the out-of-plane components is negligible for $b=10$ and $b=2.5$ and still small in case of $b=0.25$. This is because for the out-of-plane tensor elements $\chi_{xzx}^{(2)}$, $\chi_{zxx}^{(2)}$, $\chi_{yzy}^{(2)}$ and $\chi_{zyy}^{(2)}$ either the signal or idler photon will be generated by an out-of-plane nonlinear source, leading to an emission under a large propagation angle. As a result, emission via these tensor elements can only be collected efficiently with large numerical apertures. Contrarily, the tensor elements that generate the entangled states $\chi_{yyy}^{(2)} = -\chi_{yxx}^{(2)} = -\chi_{xxy}^{(2)} = -\chi_{xyx}^{(2)}$ only have in-plane components, such that the emission is close to the optical axis. The difference is clearly seen in Fig. S4, where we compare the calculated farfield emission patterns for the in-plane and out-of-plane components. While emission from the in-plane elements is concentrated around the z-axis (Fig. S4(a)), the out-of-plane components have zero emission along the z-axis (Fig S4(b)).

This means that irrespective of the ratio between d_{16}/d_{31} , the quantum state emitted along the z-axis will always only be generated by in-plane tensor components. Since the relation of the in-plane elements $\chi_{yyy}^{(2)} = -\chi_{yxx}^{(2)} = -\chi_{xxy}^{(2)} = -\chi_{xyx}^{(2)}$ directly stems from the crystal symmetry, it is not subject to dispersion. With this, the quality of the entangled state generated by 3R-MoS₂ along the z-axis does not depend on the ratio of the in- and out-of-plane tensor components which could change with wavelength. This is true even in an extreme case, where the out-of-plane elements are nominally much stronger than the in-plane tensor elements. The degree of entanglement of photon pairs emitted close to the z-axis is the practically important configuration, since the use of longer crystals together with quasi-phasematching, cavities etc. will naturally lead to directed emission close to the z-axis. However, in this work we show experimentally, that also for collection with a moderate numerical aperture of $NA=0.4$ in practice a highly entangled state can be observed.

Furthermore, we also formulate the discussion in the main text more precisely:

The out-of-plane, z-polarized nonlinear tensor components of 3R-MoS₂ practically only contribute to the generated quantum state for very large collection numerical apertures, refer to supplementary note S2 for a detailed discussion.

Taking the ratio of in-plane and out-of-plane nonlinear tensor components measured [14] as reference, these calculations predict the generation of ideal Bell states, compare the theoretical density matrices in Fig. 4(e-f).

Some other (minor) points:

a) It would be good to mention that concurrence and fidelity are defined in the Supplementary Information right after these parameters are first mentioned in the main text. This information is currently given a bit too late.

We implemented this and refer to the definitions in supplementary section now right after the first mentioning of the quantity 'fidelity' in the results section. While we also use the term 'fidelity' in the introduction, we have the impression that this would not yet be a suitable place to refer to very technical details related to the definition of some quantities.

b) Just as a reference to readers, I suggest that the authors mention the coherence length between pump, signal and idler in the main text.

This is a good suggestion, which we implemented by adding the following statement to the main text: **We however note, that this crystal thickness does not correspond to the global maximum of conversion efficiency. This would be reached for a thickness of $\approx 800 \text{ nm}$, close to the coherence length of $L_c \approx 840 \text{ nm}$ for the SHG conversion process excited at 1576 nm .**

We have detailed the calculation of the coherence length in the answer to major revision 1 of reviewer #1.

c) The green area in Fig. 2b should be explained in the figure caption.

That is a good point, we added the corresponding explanation:

The green shaded area marks the region where no photoluminescence signal distinguishable from the detector darkcounts can be detected.

d) In the second paragraph of section I-C, the parameter t (in " $t=285 \text{ nm}$ ") appears without ever being defined. I suppose that this is the crystal thickness, but this should be explicitly defined.

This is correct, we did use the caption "thickness t " in the caption of Figure 1d, however unfortunately missed to define it in the main text. We wrote now "**thickness $t=285 \text{ nm}$** ".

Reviewer #3 (Remarks to the Author):

In the manuscript by M.A. Weissflog et al., the authors demonstrate the generation of polarization-entangled photon pairs through a submicron transition metal dichalcogenide (TDM) crystal. The work presented in this manuscript pioneers the demonstration of spontaneous parametric down conversion (SPDC) process in TDM crystals, while previous literature has been limited to second harmonic generation (SHG). The authors claim and experimentally show the realization of maximally entangled Bell states, specifically $|\Phi^-\rangle$ and $|\Psi^+\rangle$, which can be respectively obtained by pumping the crystal with a y-polarization and x-polarization pump. By pumping the crystal with a ϕ_p -rotated pump polarization, the authors show that a maximally entangled state in a superposition of $|\Phi^-\rangle$ and $|\Psi^+\rangle$ can be generated, as well, with same fidelity. The demonstration of maximal entanglement is proved through concurrence estimation (described in all its details in the Supplementary Information), theoretically for all states ($|\Phi^-\rangle$, $|\Psi^+\rangle$, and their superposition) and experimentally for the $|\Phi^-\rangle$ and $|\Psi^+\rangle$ states. The authors further measure quantum state tomography and fidelity to validate the generation of states $|\Phi^-\rangle$ and $|\Psi^+\rangle$. Coincidence-to-accidental ratio is estimated to validate the quantum sources at different pump powers, while the Green's function quantization approach is theoretically utilized to describe photon pair generation.

This work can have a considerable impact in quantum photonics, especially with respect to the development of integrable entangled photon sources utilizing SPDC. In turn, the reported results can bring contribution to quantum technologies relying on highly efficient, bright, integrated sources of entangled photons. Considering these aspects and the importance of the work for specialized fields, I recommend the manuscript by M.A. Weissflog et al. for publication to Nature Communications.

Some relatively minor comments and questions should be considered before submission, which are listed in the following.

- The entangled state in Eq. (1) of the main text is a superposition of two Bell states. It is a maximally entangled state while it does not belong to multipartite entangled states (e.g., genuine multipartite or hyperentangled states). Is there a targeted applications the authors have in mind for such a state? Is it more convenient to select precise angles ϕ_p or set them to 0° or 90° to obtain the standard Bell states?

We thank the reviewer for this comment. Indeed, the generated states are bipartite states, the generation of hyperentanglement is not analyzed or demonstrated. Whereas multipartite states are highly desirable, also bipartite states, particularly the maximally entangled Bell states as demonstrated in our experiments, have several applications. For instance, they are used in many demonstrations of quantum communication protocols, including of communication using low-earth-orbit satellites [J. Yin et al., "Satellite-based entanglement distribution over 1200 kilometers" Science 356, 1140 (2017). Ref. 9 in main text]. In this application demonstration, an interferometer-based entanglement source was deployed on the satellite, which consists of a rather complex optical arrangement. Using our approach, provided that the achievable photon-pair rates can be significantly increased, such sources could be realized in a more compact and lightweight way, thus benefitting the widespread application of quantum communication.

In our source, the degree of entanglement does actually not depend on the exact pump polarization angle in the sense that any excitation angle will lead to a maximally entangled state.

It now depends on the application, for instance the particular communication protocol used, if the standard Bell states obtained for excitation along the 0° or 90° axis or any other angle are more convenient to be used. Once the crystal angle of the used MoS_2 crystal is determined, any pump angle can be used, where the accuracy for setting the pump polarization angle is the same in any case.

- With respect to this, the quantum state in Eq. (1) is only reported theoretically. Is there any experimental limitation for not demonstrating a state with a value of ϕ_p which is not either 0° or 90° and evaluating quantum state tomography and concurrence (measured experimentally)?

As stated above, no experimental limitation exists with respect to setting the pump polarization angle. We chose to demonstrate the generation of the standard Bell states, as these are commonly used in applications like quantum communication.

- On line 116 of the main text, the authors write that they “choose an area far away from the crystal edges and all cracks”. Was there a quantitative parameter/criterion for choosing such an area?

We determine the measurement position here in a two-step process, based on optical inspection of the sample and scanning SHG imaging of the entire crystal. The reason to stay away from any crystal edges, the edges of different “plateaus” of the crystal structure or obvious cracks is to avoid any regions where strain in the crystal might distort the nonlinear tensor (which would deteriorate the quality of entanglement there). This effect of distortion of the nonlinear tensor by strain has been observed in several previous works, e.g. in L. Mennel, et. al. , APL Photonics, 4, 3, p. 034404, (2019). (Ref. 32 in main text), or S. Chang et al., Nanotechnology, 35, 14, p. 145201, (2024). In the latter work it was shown that the effect of strain on the nonlinear tensor is rather localized around defect centers, which were deliberately introduced by nanoparticles distorting the TMD. This was probed by the spatial homogeneity of the SHG response of the TMD. For our work we therefore drew the conclusion that that the quantitative criterion for choosing the measurement position is keep a distance of several micrometer from any visible defects to avoid distortions of the nonlinear tensor. We then further checked that there is no strong variation in SHG intensity visible in the measurement region. Within regions that satisfy these criteria, we can freely choose a measurement position.

- In line 135, the authors write that a “ $\text{CAR} > 2$ together with the linear scaling of the coincidence rate of the coincidence rate with the pump power is clear evidence for the SPDC origin of the coincidence peak”. Why exactly 2? Is this something related to the source or the pumping scheme?

The reviewer is correct, the criterion relates to the source type which emits pairs of photons, i.e. “bunched” emission”. However, we have to thank the reviewer for raising this question, since it made us realize that we misplaced the number in this inequality. What we intended to write was that the coincidence-to-accidental ratio CAR exceeds 1. This statement is equivalent to the second-order correlation function at zero time-delay exceeding 2, i.e. $g^{(2)}(0) > 2$. Both quantities, the coincidence-to-accidental ratio CAR and $g^{(2)}(0)$ are very related. CAR is defined as $\text{CAR} = \frac{R_{\text{SPDC}}}{R_{\text{acc}}} = \frac{R_c - R_{\text{acc}}}{R_{\text{acc}}}$ where the total coincidence rate R_c is corrected for the accidental coincidence rate R_{acc} to obtain the SPDC rate R_{SPDC} . The normalized second-order correlation function $g^{(2)}(0)$ can be measured as $g^{(2)}(0) = \frac{R_c}{R_{\text{acc}}}$. Therefore, the relation between both quantities is simply $\text{CAR} = g^{(2)}(0) - 1$ (compare e.g. C. Okoth, et. al., Phys. Rev. Lett., 123, 26, p. 263602, (2019), 24 in main text).

The meaning of the statement is the same, however, to avoid confusion we mention now both versions:

A normalized second-order correlation function $g^{(2)}(0) > 2$, related to CAR via $CAR = g^{(2)}(0) - 1$ [24], together with the linear scaling of the coincidence rate with pump power shown in Fig. 2(c) is clear evidence for the SPDC origin of the coincidence peak (see supplementary note S5.A and raw data in supplementary Fig. S7)

In general, this statement is rooted in the statistics of different kind of light sources. For instance, a coherent source like a laser will show intensity correlations of $g^{(2)}(0) = 1$, while a classical, chaotic (thermal) light source shows intensity correlations of $g^{(2)}(0) = 2$ (compare R. Loudon, The Quantum Theory of Light, 3rd ed. Oxford University Press, 2000). Therefore, a light-source with a $g^{(2)}(0)$ that largely exceeds the value of 2, i.e. shows strong photon bunching, cannot be of thermal nature. Combined with the linear power scaling of the coincidence rate with pump power, this is evidence for a photon-pair source, in our case based on SPDC.

To make the definitions clearer, we also add the following paragraph to supplementary note S5.A which shows already the raw data for the CAR measurement:

The coincidence-to-accidental ratio (CAR) follows from the ratio of SPDC rate R_{SPDC} to the accidental coincidence rate R_{acc} . The SPDC rate is the measured total coincidence rate R_c corrected for accidental coincidences $R_{SPDC} = R_c - R_{acc}$. A very related quantity is the normalized second-order correlation function at zero time delay $g^{(2)}(0)$ which can be measured as $g^{(2)}(0) = \frac{R_c}{R_{acc}}$. Therefore we have in total the relation [12]

$$CAR = \frac{R_{SPDC}}{R_{acc}} = \frac{R_c - R_{acc}}{R_{acc}} = g^{(2)}(0) - 1.$$

- In the CAR plot in Fig. S3 of the Supplementary Information, the car value at 5 mW seems a bit large compared to other pump powers, unless I am missing something.

The reviewer is correct, the CAR value at 5 mW is indeed larger than for the measurements with the higher pump powers. This is in general expected, since CAR, and the related second-order correlation function at zero time-delay $g^{(2)}(0)$, are inversely proportional to the pump power P , i.e. $CAR \propto \frac{1}{P}$, compare for instance C. Okoth, et. al., Phys. Rev. Lett., 123, 26, p. 263602, (2019), 24 in main text). In our source, this inverse power dependence is actually not ideally observed, as was remarked by reviewer #1 in major revision 5. We attribute this to the relatively high noise level in the current source configuration, please see the answer to reviewer #1, major revision 5 for a more detailed discussion.

REVIEWERS' COMMENTS

Reviewer #1 (Remarks to the Author):

In the revised version of the manuscript Weissflog and co-authors have thoroughly addressed all my questions and comments, and they have satisfactorily implemented all the changes required.

As a side note, their response to my feedback stands out as one of the most comprehensive and detailed responses I have ever received.

Given the thoroughness of their revisions and the high quality of the manuscript, I am confident in recommending its publication in Nature Communications.

Reviewer #2 (Remarks to the Author):

I believe the authors have satisfactorily addressed the concerns of all three reviewers and am happy to recommend the manuscript's publication in Nature Communications. In fact, their rebuttal letter is extremely instructive on its own. I am glad that it will be available for the general public, once the article is published.

Reviewer #3 (Remarks to the Author):

After a careful reading of the answers provided by the authors, as well as of the revised manuscript, I can state that they have carefully and successfully addressed the comments that I have risen in my report. In my opinion, they also have satisfactorily replied the major and minor concerns that the other reviewers pointed out. The careful revision from the coauthors made the revised manuscript way more complete, well written, and clear than the previously submitted version. In view of these aspects, I can recommend the manuscript titled "A Tunable Transition Metal Dichalcogenide Entangled Photon-Pair Source" for publication to Nature Communications.